# An Improved Clique-Picking Algorithm for Counting Markov Equivalent DAGs via Super Cliques Transfer

**Lifu Liu** [* 1]  **Shiyuan He** [* 2]  **Jianhua Guo** [2]

## Abstract

Efficiently counting Markov equivalent directed acyclic graphs (DAGs) is crucial in graphical causal analysis. Wienöbst et al. (2023) introduced a polynomial-time algorithm, known as the Clique-Picking algorithm, to count the number of Markov equivalent DAGs for a given completed partially directed acyclic graph (CPDAG). This algorithm iteratively selects a root clique, determines fixed orientations with outgoing edges from the clique, and generates the unresolved undirected connected components (UCCGs). In this work, we propose a more efficient approach to UCCG generation by utilizing previously computed results for different root cliques. Our method introduces the concept of super cliques within rooted clique trees, enabling their efficient transfer between trees with different root cliques. The proposed algorithm effectively reduces the computational complexity of the Clique-Picking method, particularly when the number of cliques is substantially smaller than the number of vertices and edges.

## 1. Introduction

Directed acyclic graphs (DAGs) are widely used to represent multivariate causal structures across diverse fields, including epidemiology, biology, and economics (Pearl, 1988; Pingault et al., 2018). In a DAG, nodes represent variables, and directed edges denote causal relationships (Koller & Friedman, 2009; Spirtes et al., 2001). Under the Markov condition and faithfulness assumption, the causal structure can be inferred from statistical data to identify a DAG. The

d-separation properties of the identified DAG correspond to the conditional independencies observed in the data (Pearl, 2009; Verma & Pearl, 1992; Spirtes et al., 2001). However, observational data alone is often insufficient to uniquely determine the true DAG. Instead, it can identify a set of DAGs that encode the same conditional independencies, collectively known as a Markov equivalence class (MEC). This limitation has driven extensive research into learning MECs from both observational and interventional data (Perlman, 2001; Geiger & Heckerman, 2002; Chickering, 2002; Castelo & Perlman, 2004; Maathuis et al., 2009).

A MEC can be uniquely represented by an essential graph (Andersson et al., 1997), also known as the completed partially directed acyclic graph (CPDAG). They use both directed and undirected edges to represent causal relationships that are consistent across all DAGs in the equivalence class, with directed edges indicating fixed causal directions and undirected edges reflecting ambiguous dependencies unresolved by conditional independence constraints. The size of a MEC, defined as the number of DAGs within the class, plays a critical role in the design of causal intervention experiments (He & Geng, 2008) and average causal effect estimation (Maathuis et al., 2009).

Exhaustive search for all Markov equivalent DAGs is only computationally feasible for small graphs (Madigan et al., 1996; Gillispie & Perlman, 2002). Generally, the size of a MEC grows superexponentially in the number of its vertices. He et al. (2015) addressed the counting challenge by introducing five special MECs with explicit size formulas, and exploiting recursive partitioning into the respective subclasses for efficient counting. A modified approach by Ghassami et al. (2019) leverages the clique tree representation to decompose the essential graph into smaller components. More recently, dynamic programming enhancements (Ganian et al., 2022) and iterative methods over possible interventional essential graphs (AhmadiTeshnizi et al., 2020) have been proposed.

Notably, Wienöbst et al. (2023) introduces the Clique-Picking (CP) algorithm, which is a polynomial-time algorithm for determining the size of a MEC. This method partitions the MEC into subclasses by fixing a clique as a root and avoids overcounting using minimal separators

---

[*]Equal contribution  [1]School of Mathematics and Statistics, Northeast Normal University, Changchun, China  [2]School of Mathematics and Statistics, Beijing Technology and Business University, Beijing, China. Correspondence to: Jianhua Guo <jhguo@btbu.edu.cn>.

*Proceedings of the $42^{nd}$ International Conference on Machine Learning*, Vancouver, Canada. PMLR 267, 2025. Copyright 2025 by the author(s).

derived from the clique tree representation. However, for a given chordal graph $G = (V, E)$, the algorithm needs to recursively select a clique as root, introduce outgoing edges from the root, and determine smaller undirected connected components (UCCG) from the resulting graph. Suppose there are $m$ maximal cliques in $G$, this intensive process has a cost of $\mathcal{O}(m(|V| + |E|))$. Moreover, this process has to be repeated during the recursive function calls, until the reduced UCCG only contains a single maximal clique.

Fortunately, an improvement is feasible because there are considerable structure overlaps when different cliques are selected as the roots. We propose a novel approach for this purpose. Our contributions are summarized as follows.

1. We introduce higher level structures called super clique and super residual for a clique tree in Section 5. For a chordal graph $G$ with some clique selected as the root, we show the UCCGs can be easily identified from the super residuals.

2. When two different cliques $(K_i, K_j)$ are selected the root, we show structure changes can be easily identified for the corresponding super cliques and super residuals. Hence, when the UCCGs are known for $K_i$ as the root, we can efficiently identify UCCGs for the case when the other $K_j$ is the root. This leads to the super clique transfer operation in Section 6.

3. The above techniques lead to the Super Cliques Transfer Algorithm in Section 4. Overall, our procedure of UCCG identification for all root cliques has a reduced cost of $\mathcal{O}(m^2)$.

4. To provide a solid theoretical foundation for our algorithm, we characterize super cliques and super residuals from two distinct perspectives: the clique-rooted tree perspective in the main text and the clique sequence perspective in the Appendix. The former offers an intuitive understanding, while the latter provides a more fundamental framework that facilitates theoretical proofs.

The rest of the paper is as follows. Section 2 reviews the concepts for Markov equivalent DAGs and clique rooted trees. Section 3 reviews the Clique-Picking algorithm of Wienöbst et al. (2023) at a high level. Our proposed algorithm and its detailed operations will be presented in Sections 4–6. Section 7 presents the experimental results.

## 2. Preliminaries

### 2.1. Markov equivalent DAGs

A graph $G = (V, E)$ is a tuple consisting of a vertex set $V = \{v_1, \cdots, v_n\}$ and an edge set $E$. An edge $v_i - v_j$ is *undirected* if $(v_i, v_j), (v_j, v_i) \in E$ and *directed* $v_i \rightarrow v_j$ if $(v_i, v_j) \in E$ and $(v_j, v_i) \notin E$. We denote the *induced*

*subgraph* of $G$ on a set $C \subseteq V$ by $G[C]$, which only keeps the vertices in $C$ and the egdes connecting them. A *directed acyclic graph* (DAG) is a directed graph without any directed cycle. A topological ordering of a DAG is a linear ordering of its vertices such that for every directed edge $v_i \rightarrow v_j$, vertex $v_i$ appears before vertex $v_j$ in the ordering. The *skeleton* of a graph $G$ is the undirected graph formed by ignoring the edge directions in $G$, while retaining its vertices and edges. An induced subgraph of the form $v_1 \rightarrow v_2 \leftarrow v_3$ is a *v-structure*.

A *Markov equivalence class* (MEC) is the set of all DAGs that encode the same conditional independence relations among the variables. Verma & Pearl (1990) state that two DAGs are markov equivalent if and only if they share the same skeleton and $v$-structures. Furthermore, Andersson et al. (1997) show that a MEC can be uniquely represented by a completed partially directed acyclic graph (CPDAG), denoted as $G^*$, which is the union of all DAGs in the equivalence class. An undirected graph is *chordal* if every cycle of length greater than three has a chord, i.e., an edge connecting two nonconsecutive vertices in the cycle. Each *undirected connected component* of a CPDAG is a connected chordal graph, referred to as a UCCG. In particular, each UCCG is itself a CPDAG representing a MEC.

Let $\text{Size}(G^*)$ denote the size of the Markov equivalence class represented by a CPDAG $G^*$. The value of $\text{Size}(G^*)$ equals the product of the number of Markov equivalent DAGs for each UCCG of $G^*$ (Andersson et al., 1997):

$$\text{Size}(G^*) = \prod_{\text{UCCG } G \text{ in } G^*} \text{Size}(G).$$

However, the above equation is not directly applicable in general to compute $\text{Size}(G^*)$. This is because the value of $\text{Size}(G)$ can grow superexponentially with respect to its vertex number $|V|$. It is essential to develop an efficient approach for the computation of $\text{Size}(G)$.

### 2.2. Clique Rooted Trees

In a graph, a clique is a set of pairwise adjacent vertices. For a UCCG $G$, we denote the set of all its maximal cliques as $\mathcal{K}_G = \{K_1, \ldots, K_m\}$. For example, the chordal graph $G$ in Figure 1(a) contains seven cliques: $K_1 = \{a, b, c\}$, $K_2 = \{b, c, d\}$, $K_3 = \{b, e\}$, $K_4 = \{e, f\}$, $K_5 = \{b, g, j\}$, $K_6 = \{b, g, i\}$ and $K_7 = \{b, h, j\}$. The maximal cliques in a chordal graph $G$ can be ordered to satisfy the running intersection property (RIP, Blair & Peyton, 1993).

**Definition 2.1.** (Running intersection property) A clique sequence, $K_1, K_2, \ldots, K_m$, has the running intersection property (RIP) if for each clique $K_p$ (with $p = 2, \ldots, m$), there exists a clique $K_t$ for some $t \in \{1, \ldots, p-1\}$, such that

$$K_p \cap (K_1 \cup K_2 \cup \cdots \cup K_{p-1}) \subset K_t. \tag{1}$$

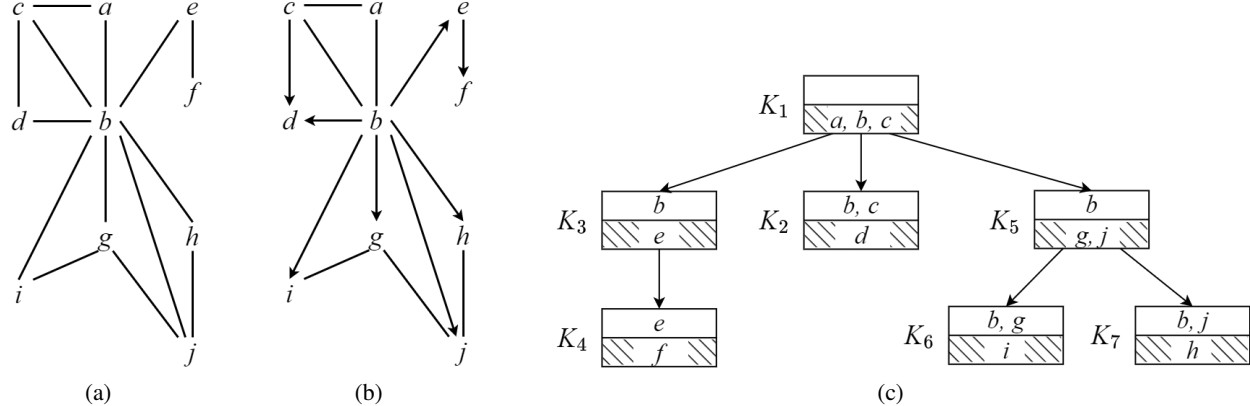

*Figure 1.* The UCCG $G$ in Figure 1(a) has several maximal cliques: $K_1 = \{a, b, c\}$, $K_2 = \{b, c, d\}$, $K_3 = \{b, e\}$, $K_4 = \{e, f\}$, $K_5 = \{b, g, j\}$, $K_6 = \{b, g, i\}$ and $K_7 = \{b, h, j\}$. Figure 1(b) is the $G^{K_1}$ with $K_1 = \{a, b, c\}$. The rooted clique tree $T^{K_1}$ is shown in Figure 1(c).

From any sequence of cliques satisfying RIP, we construct a rooted tree on $\mathcal{K}_G$ by making each clique $K_p$ adjacent to a "parent" clique $K_t$ in (1). This tree has the first clique $K_1$ in the sequence as its root, and is denoted as $T^{K_1}$. For the example in Figure 1(a), $K_1, K_2, K_3, K_4, K_5, K_6, K_7$ is an RIP sequence. The corresponding rooted tree $T^{K_1}$ is shown in Figure 1(c).

Given a UCCG $G$, a clique tree $T^{K_1}$ with some root clique $K_1$ can be generated via the MCS algorithm (Blair & Peyton, 1993). For each clique $K_p$, its separator $S_p$ is as $S_p = K_p \cap (K_1 \cup \cdots \cup K_{p-1}) = K_p \cap K_t$, and its residual $R_p$ is defined as $R_p = K_p \backslash S_p$. For the root clique, we simply set $S_1 = \emptyset$ and $R_1 = K_1$. For the rooted clique tree $T^{K_1}$, the collection of all separators is denoted as $\text{Sep}(K_1)$, and all residuals as $\text{Res}(K_1)$.

## 3. The Clique-Picking Algorithm

We now review the Clique-Picking algorithm proposed by Wienöbst et al. (2023) and highlight, at the end of this section, the specific part where our super clique approach can provide improvements. Wienöbst et al. (2023) exploited the fact that each DAG within MEC can be represented by topological vertex orderings, and a maximal clique can be selected as the prefix of an ordering. In this way, the Markov equivalent DAGs can be divided into small groups for more efficient computation.

Wienöbst et al. (2023) introduced several concepts to formalize the idea. Suppose $K$ be a clique in $G$ and selected as the root. Let $\pi(K)$ be a permuted ordering of the vertices in $K$, and consider all topological orderings of $G$ that start with $\pi(K)$. The $\pi(K)$-orientation of $G$, denoted $G^{\pi(K)}$, is the union of all DAGs within the MEC represented by $G$ that have topological orderings beginning with $\pi(K)$. Then,

**Algorithm 1** Function CP-Count($\cdot$)

**Input:** A UCCG $G$
**Output:** Size($G$)
1: Generate a rooted clique tree of $G$;
2: Generate $\mathcal{C}_G(K_p)$ for each $K_p \in \mathcal{K}_G$, which is selected as the root for clique tree $T^{K_p}$;
3: Evaluate Size($J$) for all UCCG $J$ inside $\mathcal{C}_G(K_p)$ by recursively calling CP-Count($J$);
4: Compute Size($G$) in (2).

denote by $\mathcal{C}_G(\pi(K))$ the undirected connected components of $G^{\pi(K)}[V \setminus K]$.

Furthermore, let $G^K$ denote the union of $\pi(K)$-orientations of $G$ over all permutation $\pi$. That is, $G^K = \bigcup_\pi G^{\pi(K)}$. We also denote $\mathcal{C}_G(K)$ as the undirected connected components of $G^K[V \setminus K]$. For the graph in Figure 1(a), suppose $K_1 = \{a, b, c\}$ is picked as the prefix of the ordering, then the corresponding graphs $G^{K_1}$ is shown in Figure 1(b). We can see, by picking $K_1$ as the root, we introduce outgoing edges from $K_1$ in $G^{K_1}$, compared with the original undirected $G$ in Figure 1(a). For $G^{K_1}$, we have the undirected connected components $\mathcal{C}_G(K_1) = \{G[e], G[d], G[f], G[g, h, i, j]\}$.

Wienöbst et al. (2023) show that the size of the Markov equivalence class represented by $G^K$ can be calculated by:

$$\text{Size}(G^K) = |K|! \cdot \prod_{J \in \mathcal{C}_G(K)} \text{Size}(J).$$

It is tempting to select each $K$ in $\mathcal{K}_G$, compute Size($G^K$) and sum all these values to get Size($G$) for a UCCG $G$. However, this will count some DAGs multiple times, as a DAG can be represented by multiple topological orderings with different cliques as the root. To resolve this issue,

---

**Algorithm 2** Super Cliques Transfer Algorithm

---

**Input:** A UCCG $G$ and a rooted clique tree $T^{K_1}$ of $\mathcal{K}_G$
**Output:** $\mathcal{C}_G(K_1), \mathcal{C}_G(K_2), \ldots, \mathcal{C}_G(K_m)$.
1: $L^{(1)}, \mathcal{C}_G(K_1), \text{Sep}(K_1), \text{Res}(K_1) \leftarrow \text{SC-Create-Op}$
    $(G, T^{K_1})$ via Algorithm 3;
2: **for** $i = 2 \text{ to } m$ **do**
3:     $K_t \leftarrow$ The parent clique of $K_i$ in $T^{K_1}$;
4:     Initialize $\text{Sep}(K_i) \leftarrow \text{Sep}(K_t)$ and $\text{Res}(K_i) \leftarrow \text{Res}(K_t)$;
5:     Update $S_i \leftarrow \emptyset$ and $S_t \leftarrow K_i \cap K_t$ in $\text{Sep}(K_i)$;
6:     Update $R_i \leftarrow K_i$ and $R_t \leftarrow K_t \setminus (K_i \cap K_t)$ in $\text{Res}(K_i)$;
7:     Run Algorithm 4 to get $\mathcal{C}_G(K_i), L^{(i)} \leftarrow$
        $\text{SC-Trans-Op}(\mathcal{C}_G(K_t), L^{(t)}, T^{K_t}, \text{Sep}(K_i), \text{Res}(K_i))$;
8:     Get $T^{K_i}$ by reversing the edge "$K_t \rightarrow K_i$" in $T^{K_t}$ to
        "$K_i \rightarrow K_t$";
9: **end for**

---

**Algorithm 3** SC-Create-Op

---

**Input:** A UCCG $G$, a rooted clique tree $T^{K_1}$ of $\mathcal{K}_G$.
**Output:** $L^{(1)}, \mathcal{C}_G(K_1), \text{Sep}(K_1)$, and $\text{Res}(K_1)$.
1: Initialize $\mathcal{C}_G(K_1) \leftarrow \{\}$;
2: $\text{Sep}(K_1) \leftarrow$ the set of separators $S_1, \ldots, S_m$;
3: $\text{Res}(K_1) \leftarrow$ the set of residuals $R_1, \ldots, R_m$;
4: Based on $\text{Sep}(K_1)$, get the set of super cliques of $T^{K_1}$ and
    denote it as $L^{(1)}$;
5: **for** $\text{SK}_{p^+}^{(1)}$ in $L^{(1)}$ **do**
6:     Obtain $\text{SR}_{p^+}^{(1)}$ for $\text{SK}_{p^+}^{(1)}$ based on $\text{Res}(K_1)$;
7:     $\mathcal{C}_G(K_1) \leftarrow \mathcal{C}_G(K_1) \cup \{G[\text{SR}_{p^+}^{(1)}]\}$.
8: **end for**

---

Wienöbst et al. (2023) further introduced the correct iterative formula

$$\text{Size}(G) = \sum_{p=1}^{m} \phi(K_p, \text{FP}(K_p, T^{K_1})) \cdot \prod_{J \in \mathcal{C}_G(K_p)} \text{Size}(J), \quad (2)$$

where $\phi(\cdot)$ is a corrected multiplicative factor to avoid overcounting. The formal definition of the above $\phi(\cdot)$ is provided in Section 4.3 of Wienöbst et al. (2023), where the authors discuss it in detail.

At a high level, the recursive strategy of Wienöbst et al. (2023) is summarized in Algorithm 1. We present their procedure as a function, called CP-Count($\cdot$), that takes a UCCG $G$ as input and return the number of Markov equivalent DAGs, i.e. $\text{Size}(G)$. In particular, Step 2 of Algorithm 1 generates all $\mathcal{C}_G(K_p)$, when different $K_p$'s are selected as the root. Wienöbst et al. (2023) proposed an adapted Maximum Label Search algorithm for this task. The cost of generating a single $\mathcal{C}_G(K_p)$ is $\mathcal{O}(|V| + |E|)$, and the overall cost for Step 2 of Algorithm 1 is $\mathcal{O}(m \cdot (|V| + |E|))$, where $m$ is number of maximal cliques in the chordal graph $G$.

It is important to note Step 2 of Algorithm 1 can be further improved to achieve greater efficiency. This is because there are considerable structure overlap for $\mathcal{C}_G(K)$ with different $K \in \mathcal{K}_G$ selected as the root. For example, we can easily see that, when $K_3$ is selected as the root, we have $\mathcal{C}_G(K_3) = \{G[a, c, d], G[f], G[g, h, i, j]\}$. It is clear that the undirected connected components $G[f]$ and $G[g, h, i, j]$ appear in both $\mathcal{C}_G(K_1)$ and $\mathcal{C}_G(K_3)$. To address this redundancy, we will introduce our super clique transfer algorithm in the next section. With the proposed algorithm, once we get $\mathcal{C}_G(K_1)$, we can more efficiently compute and derive all the other $\mathcal{C}_G(K_p)$ with $p = 2, \ldots, m$. The reduced cost is $\mathcal{O}(m^2)$ for Step 2 of Algorithm 1.

## 4. The Super Cliques Transfer Algorithm

Our main contribution is a novel algorithm that reduces the computation cost of Step 2 of Algorithm 1. The main idea is to group the cliques in a rooted clique tree into higher level structure, called super cliques. We connect the super cliques with the UCCG, and develop an efficient super clique transfer algorithm to obtain the UCCGs when different cliques are selected as the root.

The proposed approach, referred to as the Super Cliques Transfer (SC-Trans) Algorithm, is outlined in Algorithm 2. It takes as input a UCCG $G$ and a corresponding rooted clique tree $T^{K_1}$, and outputs all sets $\mathcal{C}_G(K_1), \ldots, \mathcal{C}_G(K_m)$. Step 1 identifies all separators $\text{Sep}(K_1)$, all residuals $\text{Res}(K_1)$, all super cliques $L^{(1)}$ and $\mathcal{C}_G(K_1)$ for $T^{K_1}$. It utilizes the super clique create operation (SC-Create-Op) in Algorithm 3, which will be introduced in details in Section 5. Steps 2–9 sequentially generate the other $\mathcal{C}_G(K_2), \ldots, \mathcal{C}_G(K_m)$. These steps depend on the technical details to be presented in Section 6. In each iteration of $i \in \{2, \ldots, m\}$, the parent clique $K_t$ of $K_i$ in $T^{K_1}$ is found. Steps 4–7 then efficiently identify structure changes from $T^{K_t}$ to $T^{K_i}$. In particular, Algorithm 4 in Step 7 is the super clique transfer operation (SC-Trans-Op) in Section 6. Step 8 then updates $T^{K_t}$ to become a rooted tree for $K_i$.

We have the following results for Algorithm 2, the proof of which can be found in Appendix E. *Proof of Theorem 4.1* and *Proof of Theorem 4.2*.

**Theorem 4.1.** *Let $G$ be a UCCG, and $T^{K_1}$ be a rooted clique tree with cliques ordered as $K_1, \ldots, K_m$ according to the MCS algorithm. Algorithm 2 will correctly return $\mathcal{C}_G(K_1), \mathcal{C}_G(K_2), \ldots, \mathcal{C}_G(K_m)$.*

**Theorem 4.2.** *Algorithm 2 runs in time $\mathcal{O}(m^2)$, where $m$ is the number of cliques of UCCG $G$.*

## 5. Super Cliques and Undirected Connected Components

In this section, we discuss the details of Algorithm 3. It is a novel approach to compute $\mathcal{C}_G(K_1)$ based on the concept of super cliques for $T^{K_1}$. The new concepts are built upon the basic rooted clique tree structures introduced in Section 2.2.

**Algorithm 4** SC-Trans-Op

**Input:** $\mathcal{C}_G(K_t), L^{(t)}, T^{K_t}, \mathrm{Sep}(K_i), \mathrm{Res}(K_i)$.
**Output:** $\mathcal{C}_G(K_i)$ and $L^{(i)}$.
1: Initialize $\mathcal{C}_G(K_i) \leftarrow \{\}, L^{(i)} \leftarrow \{\}, \mathrm{SK}_{t+}^{(i)} \leftarrow \{K_t\}$,
   $\mathrm{SR}_{t+}^{(i)}(K_i) \leftarrow \{R_t\}$
2: **for** $\mathrm{SK}_{p+}^{(t)}$ in $L^{(t)}$ **do**
3:    **if** $p = i$ **then**
4:      **for all** child clique $K_q$ of $K_i$ in $T^{K_t}$ **do**
5:        Induce $\mathrm{SK}_{q+}^{(i)}$ and $\mathrm{SR}_{q+}^{(i)}$ from $\mathrm{SK}_{i+}^{(t)}$ and $\mathrm{SR}_{i+}^{(t)}$;
6:        $\mathcal{C}_G(K_i) \leftarrow \mathcal{C}_G(K_i) \cup \{G[\mathrm{SR}_{q+}^{(i)}]\}$;
7:        $L^{(i)} \leftarrow L^{(i)} \cup \{\mathrm{SK}_{q+}^{(i)}\}$;
8:      **end for**
9:    **else if** $K_p$ is a child clique of $K_t$ in $T^{K_t}$, and $S_t$ is a proper subset of $S_p$ **then**
10:      $\mathrm{SK}_{t+}^{(i)} \leftarrow \mathrm{SK}_{t+}^{(i)} \cup \mathrm{SK}_{p+}^{(t)}, \mathrm{SR}_{t+}^{(i)} \leftarrow \mathrm{SR}_{t+}^{(i)} \cup \mathrm{SR}_{p+}^{(t)}$;
11:    **else**
12:      $\mathrm{SR}_{p+}^{(i)} \leftarrow \mathrm{SR}_{p+}^{(t)}, \mathrm{SK}_{p+}^{(i)} \leftarrow \mathrm{SK}_{p+}^{(t)}$;
13:      $\mathcal{C}_G(K_i) \leftarrow \mathcal{C}_G(K_i) \cup \{G[\mathrm{SR}_{p+}^{(i)}]\}$;
14:      $L^{(i)} \leftarrow L^{(i)} \cup \{\mathrm{SK}_{p+}^{(i)}\}$;
15:    **end if**
16: **end for**
17: $L^{(i)} \leftarrow L^{(i)} \cup \{\mathrm{SK}_{t+}^{(i)}\}$;
18: $\mathcal{C}_G(K_i) \leftarrow \mathcal{C}_G(K_i) \cup \{G[\mathrm{SR}_{t+}^{(i)}]\}$.

---

**Definition 5.1.** (Clique header, clique tail) Let $T^{K_1}$ be a rooted clique tree with RIP clique order $K_1, K_2, \ldots, K_m$.

   i. For any $p = 2, \ldots, m$, $K_p$ is a *clique header* within $T^{K_1}$ if for any ancestral clique $K_q$ of $K_p$ with $q \neq 1$, the corresponding $S_q$ is not a proper subset of $S_p$.

   ii. For $p = 2, \ldots, m$, suppose $K_p$ is a clique header within $T^{K_1}$, a descendant clique $K_q$ of $K_p$ is a *clique tail* that follows $K_p$ if $S_p \subsetneq S_q$.

Note the root $K_1$ is neither a clique header nor a clique tail, as we require $p > 1$ in the above definitions. For the example in Figure 1(c), $K_2$, $K_3$, $K_4$ and $K_5$ are clique headers within $T^{K_1}$, $K_6$ and $K_7$ are the clique tails that follows $K_5$. Using the concepts of clique header and clique tail, we can define the super clique and super residual within $T^{K_1}$.

**Definition 5.2.** (Super clique, super residual) Within a clique tree $T^{K_1}$, suppose $K_p$ is a clique header and $K_{p_1}, \ldots, K_{p_r}$ are all its clique tails.

   i. The clique set $\mathrm{SK}_{p+}^{(1)} = \mathrm{SK}_{p|p_1,\ldots,p_r}^{(1)} := \{K_p, K_{p_1}, \ldots, K_{p_r}\}$ is called a *super clique*.

   ii. The set of the residuals corresponding to the cliques within $\mathrm{SK}_{p+}^{(1)}$ is called a *super residual*, and denoted as

$$\mathrm{SR}_{p+}^{(1)} = \mathrm{SR}_{p|p_1,\ldots,p_r}^{(1)} := \{R_p, R_{p_1}, \ldots, R_{p_r}\}.$$

A clique header $K_p$ will form a super clique itself if it does not have any clique tail. For the clique tree in the left

panel of Figure 2, there are four super cliques: $\mathrm{SK}_{2|}^{(1)} = \{K_2\}$, $\mathrm{SK}_{3|}^{(1)} = \{K_3\}$, $\mathrm{SK}_{4|}^{(1)} = \{K_4\}$ and $\mathrm{SK}_{5|6,7}^{(1)} = \{K_5, K_6, K_7\}$ within $T^{K_1}$. Note the superscript "(1)" in these notations emphasizes that they are super cliques (or super residuals) within the clique tree $T^{K_1}$ rooted at $K_1$.

Regarding the super cliques in $T^{K_1}$, we can observe a few properties. Firstly, for each $\mathrm{SK}_{p+}^{(1)}$, the subgraph $T^{K_1}[\mathrm{SK}_{p+}^{(1)}]$ is connected and constitutes a subtree of $T^{K_1}$. This is because the clique tree $T^{K_1}$ generated from a UCCG $G$ satisfies the so called induced-subtree property (Blair & Peyton, 1993). The property states that, for every vertex $v \in V$ of $G$, the set of all cliques containing $v$ induces a connected subtree of $T$. Consequently, the subgraph $T^{K_1}[\mathrm{SK}_{p+}^{(1)}]$ is connected because all the cliques within $\mathrm{SK}_{p+}$ share the common node set $S_p$.

Secondly, we observe that $\mathcal{C}_G(K_1)$ can be easily obtained from the set of super residuals. Consider again the clique tree in the left panel of Figure 2 in Figure 1(c), we can see $G[\mathrm{SR}_{2|}^{(1)}] = G[d], G[\mathrm{SR}_{3|}^{(1)}] = G[e], G[\mathrm{SR}_{4|}^{(1)}] = G[f]$, and $G[\mathrm{SR}_{5|6,7}^{(1)}] = G[g, h, i, j]$ are the undirected connected components in $\mathcal{C}_G(K_1)$. In fact, this observation holds in general. For any super residual $\mathrm{SR}_{p+}^{(1)}$, the induced subgraph $G[\mathrm{SR}_{p+}^{(1)}]$ is exactly an element of the set $\mathcal{C}_G(K_1)$. Moreover, $\mathcal{C}_G(K_1)$ is just the collection of all such subgraphs induced by every super residual.

**Theorem 5.3.** *Let $T^{K_1}$ be a rooted clique tree of a chordal graph $G$ with MCS clique order $K_1, K_2, \ldots, K_m$. Then*

$$\mathcal{C}_G(K_1) = \left\{ G[\mathrm{SR}_{p+}^{(1)}] \ : \ \mathrm{SR}_{p+}^{(1)} \text{ is a super residual within } T^{K_1} \right\}.$$

For a given $T^{K_1}$, Algorithm 3 is designed to return the set $L^{(1)}$ of all super cliques, all separators and residuals, and $\mathcal{C}_G(K_1)$. Algorithm 3 is valid due to Theorem 5.3. It seems natural that, for $i = 2, \ldots, m$, we can apply the same procedure to each $T^{K_i}$ for getting the corresponding $\mathcal{C}_G(K_i)$. However, such procedure is unnecessary. Recall we have discussed that, for any pair of cliques $K_t$ and $K_i$ with $k \neq i$, there are many shared undirected connected components between $\mathcal{C}_G(K_i)$ and $\mathcal{C}_G(K_t)$. We can reuse the computation results for one rooted clique tree to speed up the computation for the other. In the next section, we will present an efficient strategy serving this purpose.

## 6. The Super Cliques Transfer Operation

We now present our efficient super clique transfer operation to generate all the other $\mathcal{C}_G(K_2), \mathcal{C}_G(K_3), \ldots, \mathcal{C}_G(K_m)$, given $\mathcal{C}_G(K_1)$. The overall iterative strategy is described in Algorithm 2, which generates $\mathcal{C}_G(K_i)$ based on $\mathcal{C}_G(K_t)$, where $K_t$ is a parent clique of $K_i$ in $T^{K_1}$.

To efficiently obtain $\mathcal{C}_G(K_i)$ from $\mathcal{C}_G(K_t)$, we need to construct an appropriate clique tree $T^{K_i}$ with minimal structure changes from $T^{K_t}$. We exploit the computed results from $T^{K_t}$ and our super clique transfer operation to reduce the computation cost. Now, without loss of generality, we discuss in details the particular situation where we transit from $T^{K_1}$ to $T^{K_i}$, where $K_i$ is a child clique of $K_1$ in $T^{K_1}$. The procedure for the other cases is similar.

Recall $T^{K_1}$ corresponds to a clique sequence $K_1, \ldots, K_m$ that satisfies the RIP. As stated in Lemma 6.1 below, there always exists a permuted sequence $K_{\sigma(1)}, \ldots, K_{\sigma(m)}$, which starts with $K_i$ and also satisfies the RIP.

**Lemma 6.1.** *(Proposition 2.4 of Leimer (1993))* *Let $K_1, \ldots, K_m$ be an RIP sequence of the clique set. For any $i = 2, \ldots, m$, there exists a permutation $\sigma$ satisfying that $\sigma(1) = i$ and $\sigma(2) = 1$, and meanwhile $K_{\sigma(1)}, \ldots, K_{\sigma(m)}$ is still an RIP sequence.*

In fact, the permuted sequence has a simple structure change. The proof of Leimer (1993) actually states the permuted indices as: 1) $\sigma(1) = i$ and $\sigma(2) = 1$; 2) for $p = 2, \ldots, i - 1$, we have $\sigma(p + 1) = p$; and 3) for $p = i + 1, \ldots, m$, we have $\sigma(p) = p$. The permuted RIP sequence has minimal change of the clique order. Based on the permuted sequence, the new clique tree $T^{K_i}$ rooted at $K_i$ can be obtained. We continue to examine the structure changes from $T^{K_t}$ to $T^{K_i}$ in more details.

## 6.1. Basic Structure Changes in the Clique Trees

To understand the structure changes for $T^{K_i}$, we first state a property regarding the permuted sequence.

**Proposition 6.2.** *Assume $K_i$ is a clique such that $K_i \cap (K_1 \cup \cdots \cup K_{i-1}) \subset K_1$. Let $K_{\sigma(1)}, \ldots, K_{\sigma(m)}$ be the permuted clique obtained by applying Lemma 6.1 with $\sigma(1) = i$. For any $p \in [m] \setminus \{1, i\}$ and any $q \in [m]$, if $K_p \cap (K_1 \cup \cdots \cup K_{p-1}) \subset K_q$, then for $p'$ and $q'$ with $p = \sigma(p')$ and $q = \sigma(q')$, it holds that*

$$K_{\sigma(p')} \cap (K_{\sigma(1)} \cup \cdots \cup K_{\sigma(p'-1)}) \subset K_{\sigma(q')}.$$

Note in the above $p \notin \{1, i\}$. Proposition 6.2 has the following implication for any $K_q$ and its child clique $K_p$ in $T^{K_1}$. Suppose the $p$-th clique $K_p$ in $T^{K_1}$ corresponds to $p'$-th clique $K_{\sigma(p')}$ in $T^{K_i}$, and suppose $K_q$ corresponds to $K_{\sigma(q')}$. We have $K_{\sigma(p')}$ is a child clique of $K_{\sigma(q')}$ in $T^{K_i}$.

The above discussion implies that, from $T^{K_1}$ to this $T^{K_i}$, only one edge changes. That is, $K_1 \to K_i$ in $T^{K_1}$ becomes $K_i \to K_1$ in $T^{K_i}$. The other edges in $T^{K_i}$ remain unchanged. Due to this edge direction change, we can see that their separators and residuals also change. The changes are summarized in Table 1.

Additionally, the separators and residuals for the other

Table 1. The separators and residuals for the cliques $K_1$ and $K_i$ within the two rooted clique trees $T^{K_1}$ and $T^{K_i}$.

|  | $K_1$ | | $K_i$ | |
|---|---|---|---|---|
|  | separator | residual | separator | residual |
| $T^{K_1}$ | $\emptyset$ | $K_1$ | $K_1 \cap K_i$ | $K_i \setminus (K_1 \cap K_i)$ |
| $T^{K_i}$ | $K_1 \cap K_i$ | $K_1 \setminus (K_1 \cap K_i)$ | $\emptyset$ | $K_i$ |

cliques remain unchanged, which is stated in the following proposition.

**Proposition 6.3.** *Assume $K_i$ is a clique such that $K_i \cap (K_1 \cup \cdots \cup K_{i-1}) \subset K_1$. Let $K_{\sigma(1)}, \ldots, K_{\sigma(m)}$ be the permutation obtained by applying Lemma 6.1 with $\sigma(1) = i$. Then for any $p$ in $[m] \setminus \{1, i\}$ and $p'$ satisfying $p = \sigma(p')$, we have $S_p = S_{\sigma(p')}$ and $R_p = R_{\sigma(p')}$.*

The proof of above two propositions are claimed in Appendix E. *Proof of Propositions 6.2 & 6.3.* The edge $K_1 \to K_i$ in $T^{K_1}$ will be redirected as $K_i \to K_1$ in $T^{K_i}$. This implies $K_1$ becomes a child clique of $K_i$ in $T^{K_i}$. As the root $K_i$ is the only ancestral clique of $K_1$ in $T^{K_i}$, $K_1$ must be a clique header within $T^{K_i}$ by Definition 5.1. The child cliques of $K_i$ in $T^{K_1}$ will all become clique headers in $T^{K_i}$. Additionally, some cliques that were headers in $T^{K_1}$ will become clique tails of $K_1$ in $T^{K_i}$. Specifically, if $K_p$ ($p \neq i$) is a child clique of $K_1$ in $T^{K_1}$, then $K_p$ is a clique header within $T^{K_1}$, but it can possibly become a clique tail within $T^{K_i}$. We need to check whether $K_1 \cap K_i$ is a proper subset of $S_p$. If this is true, $K_p$ will become a clique tail of $K_1$ in $T^{K_i}$; otherwise, $K_p$ remain a clique header in $T^{K_i}$.

For Figure 2, let us consider the structure changes from $T^{K_1}$ in the left panel to $T^{K_5}$ in the right panel. We can see that $K_6$ and $K_7$ both become clique headers within $T^{K_5}$. Furthermore, $K_1$ also becomes a clique header in $T^{K_5}$. Since $K_1 \cap K_5$ is not a proper subset of $S_3$, $K_3$ remains a clique header in $T^{K_5}$. On the other hand, $K_1 \cap K_5$ is a proper subset of $S_2$, so $K_2$ becomes a clique tail that follows $K_1$ within $T^{K_5}$. The clique $K_4$, which is not adjacent to $K_1$, remains as a clique header within $T^{K_5}$.

## 6.2. High-level Structure Changes in the Clique Trees

We can further characterize higher level structure changes from $T^{K_1}$ to $T^{K_i}$, in terms of super cliques and super residuals. In fact, all super cliques in $T^{K_i}$ can be identified from those of $T^{K_1}$. There are three cases to consider:

1. Consider the super clique $\mathrm{SK}_{i+}^{(1)}$ with clique header $K_i$ in $T^{K_1}$. Suppose $K_i$ has $h$ child clique(s) in $T^{K_1}$: $K_{p_1}, \ldots, K_{p_h}$. Then, the super clique $\mathrm{SK}_{i+}^{(1)}$ of $T^{K_1}$ get split into $h$ super clique(s) in $T^{K_i}$: $\mathrm{SK}_{p_1+}^{(i)}, \ldots, \mathrm{SK}_{p_h+}^{(i)}$. These super cliques have $K_{p_1}, \ldots, K_{p_h}$ as their clique headers, respectively. In the special case that $K_i$ has no child clique in $T^{K_1}$,

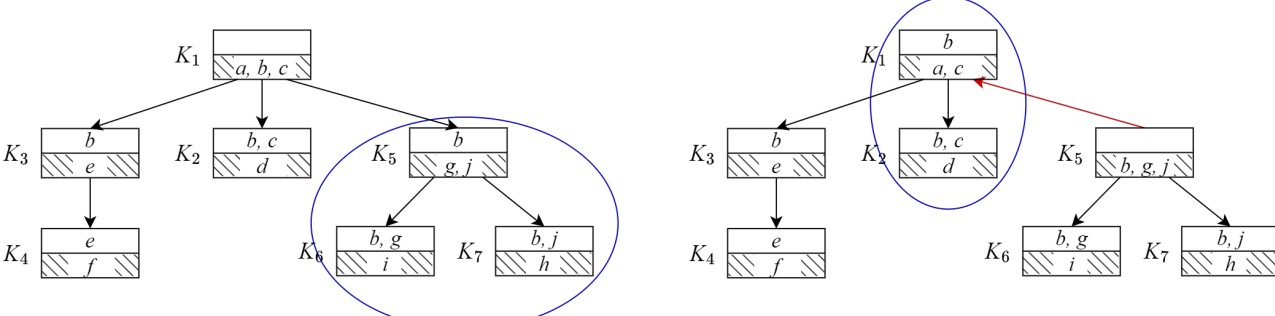

**Figure 2.** The structure changes from $T^{K_1}$ to $T^{K_5}$. The blue cycle means that the cliques within it will form a super clique. We omit the blue cycle when a clique itself is a super clique(root excluded).

we can simply ignore $\mathrm{SK}_{i+}^{(1)}$ when generating super cliques for $T^{K_i}$.

2. Consider all child cliques of $K_1$ in $T^{K_1}$ *but* with $K_i$ excluded. Among these child cliques, select those that become clique tails of $K_1$ in $T^{K_i}$, and denote these selected cliques as $K_{p_1}, \ldots, K_{p_h}$. Then, their corresponding super cliques $\mathrm{SK}_{p_1^+}^{(1)}, \ldots, \mathrm{SK}_{p_h^+}^{(1)}$ in $T^{K_1}$, together with $K_1$, will form a new super clique in $T^{K_i}$:

$$\mathrm{SK}_{1+}^{(i)} = \{K_1\} \cup \mathrm{SK}_{p_1^+}^{(1)} \cup \cdots \cup \mathrm{SK}_{p_h^+}^{(1)}. \quad (3)$$

Note, if there does not exits any clique tail of $K_1$ in $T^{K_i}$, then (3) simply becomes $\mathrm{SK}_{1+}^{(i)} = \{K_1\}$. For the child cliques of of $K_1$ in $T^{K_1}$ that are *not* selected for (3), their corresponding super cliques remain unchanged and continue to constitute super cliques in $T^{K_i}$.

3. Aside from the Case 1 and Case 2 discussed above, all other super cliques in $T^{K_1}$ remain unchanged and continue to form super cliques in $T^{K_i}$.

All super residuals in $T^{K_i}$ can be identified from those in $T^{K_1}$ in the same spirit as the three cases above. Recall by Proposition 6.3 the only difference between $\mathrm{Res}(K_i)$ and $\mathrm{Res}(K_1)$ lies in the pair $(R_1, R_i)$, and the changes are summarized in Table 1.

Corresponding to Case 1 above, the super residual $\mathrm{SR}_{i+}^{(1)}$ of $T^{K_1}$ get split into $h$ super residual(s) in $T^{K_i}$: $\mathrm{SR}_{p_1^+}^{(i)}, \ldots, \mathrm{SR}_{p_h^+}^{(i)}$. As for Case 2 in the above, the super residual $\mathrm{SR}_{1+}^{(i)}$ corresponding to $K_1$ in $T^{K_i}$ will be

$$\mathrm{SR}_{1+}^{(i)} = \{K_1 \setminus (K_1 \cap K_i)\} \cup \mathrm{SR}_{p_1^+}^{(1)} \cup \cdots \cup \mathrm{SR}_{p_h^+}^{(1)}. \quad (4)$$

Except for $\mathrm{SR}_{i+}^{(1)}$ and the super residual in (4), all the other super residuals of $T^{K_1}$ remain exactly the same in $T^{K_i}$.

Once the super residuals in $T^{K_i}$ are identified, the undirected components in $\mathcal{C}_G(K_i)$ can be immediately determined based on Theorem 5.3.

We now illustrate the structure changes from $T^{K_1}$ to $T^{K_5}$ for the example in Figure 2. Corresponding to Cases 1–3, we have the following:

1. $K_5$ has two child cliques in $T^{K_1}$: $K_6$ and $K_7$. Then the super clique $\mathrm{SK}_{5|6,7}^{(1)}$ in $T^{K_1}$ get split in two super cliques in $T^{K_5}$: $\mathrm{SK}_{6|}^{(5)}$, and $\mathrm{SK}_{7|}^{(5)}$. Correspondingly, we can generate the undirected connected components $G[i]$ and $G[h]$ in $\mathcal{C}_G(K_5)$.

2. Additionally, consider the child cliques of $K_1$ in $T^{K_1}$. In $T^{K_5}$, $K_2$ becomes a child tail of $K_1$. Then $\mathrm{SK}_{2|}^{(1)}$ in $T^{K_1}$, together with $K_1$, will form the new super cliques $\mathrm{SK}_{1|2}^{(5)}$ of $T^{K_5}$. Due to the change of residual of $K_1$, we can see $G[\mathrm{SR}_{1|2}^{(5)}] = G[a, c, d]$ is an undirected connected component in $\mathcal{C}_G(K_5)$, where $\mathrm{SR}_{1|2}^{(5)} = \{K_1 \setminus (K_1 \cap K_5)\} \cup \mathrm{SR}_{2|}^{(1)}$.

3. Except for $\mathrm{SK}_{5|6,7}^{(1)}$ and $\mathrm{SK}_{2|}^{(1)}$, all the other super cliques: $\mathrm{SK}_{3|}^{(1)}$ and $\mathrm{SK}_{4|}^{(1)}$ remain exactly the same in $T^{K_5}$, and the same apply to $\mathrm{SR}_{3|}^{(1)}$ and $\mathrm{SR}_{4|}^{(1)}$. Hence we have $G[e]$ and $G[f]$ in $\mathcal{C}_G(K_1)$ still belong to $\mathcal{C}_G(K_5)$.

### 6.3. The Iterative Algorithm

In the above, we focus on the case where $K_i$ is one child clique of $K_1$ in $T^{K_1}$, and we identify all super cliques and super residuals of $T^{K_i}$ from those of $T^{K_1}$. The results can be easily generalized. For any clique $K_i$ in $\mathcal{K}_G$ with $1 < i \leq m$, suppose its parent clique of $K_i$ in $T^{K_1}$ as $K_t$, we can efficiently obtain super cliques and super residuals of $T^{K_i}$ from those of $T^{K_t}$. This leads to the super clique transfer operation (SC-Trans-Op) in Algorithm 4, which derives the set of super cliques of $T^{K_i}$ from that of $T^{K_t}$,

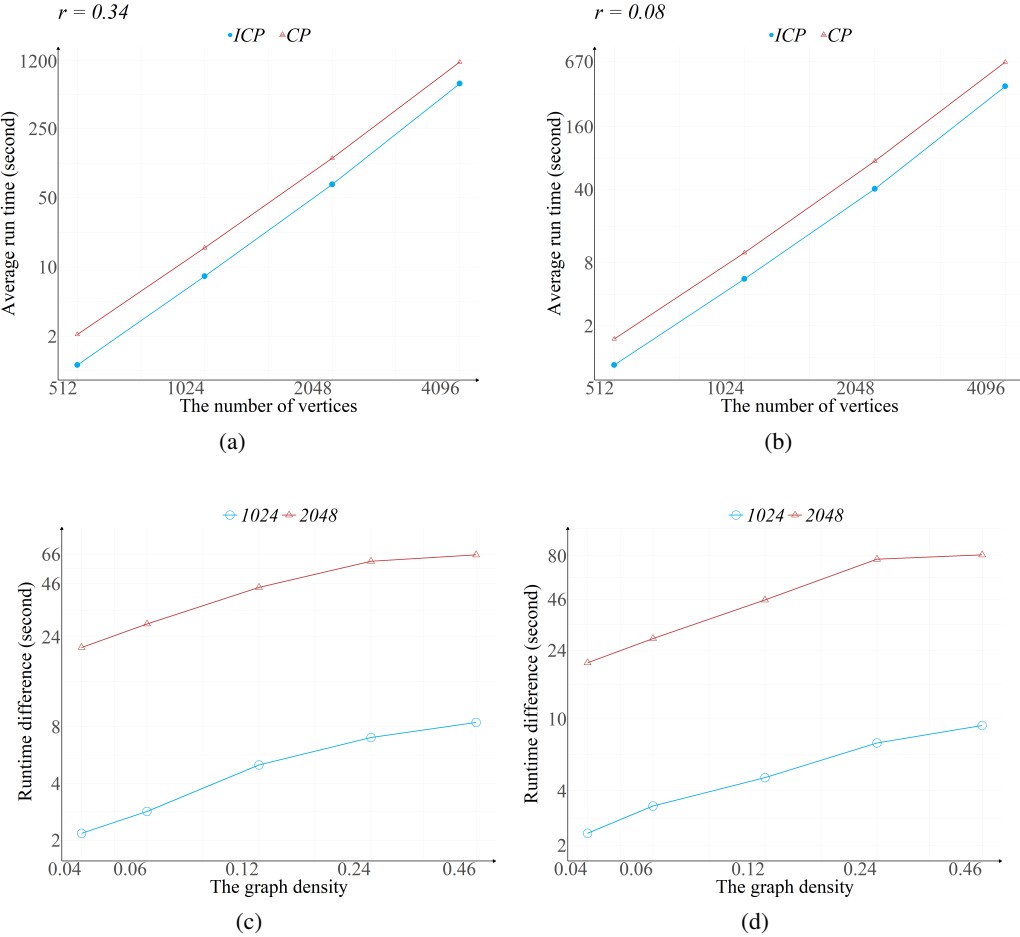

*Figure 3.* Figures (a) and (b) present the average running times of the original "CP" method and our improved "ICP" method across different numbers of graph vertices. The runtime differences, $T_{\mathrm{CP}} - T_{\mathrm{ICP}}$ and $T_{\mathrm{CP},2} - T_{\mathrm{ICP},2}$ are illustrated in Figures (c) and (d), respectively, for varying graph densities. All axes use a logarithmic scale.

and generates $\mathcal{C}_G(K_i)$ from $\mathcal{C}_G(K_t)$. In Algorithm 4, Lines 3-8, Lines 9-10 & 17, and Lines 11-15 correspond to Cases 1–3 in Section 6.2, respectively.

Now, let us return to Algorithm 2. In Algorithm 2, its Lines 4–6 and Line 8 are due to the basic structure changes in Section 6.1. When transitioning from $T^{K_t}$ to $T^{K_i}$, the set of separators ($\mathrm{Sep}(K_i)$) and the set of residuals ($\mathrm{Res}(K_i)$) can be readily determined, as we discussed for Table 1 and Proposition 6.3. Line 8 is due to Proposition 6.2 and the subsequent discussion there. Line 7 employs Algorithm 4.

In the following theorem, we prove the correctness of Algorithm 4.

**Theorem 6.4.** *For any $i = 2, \ldots, m$, let $K_t$ denote the parent clique of $K_i$ in $T^{K_1}$, $L^{(t)}$ be the set of super cliques of $T^{K_t}$. Then given $\mathcal{C}_G(K_t), L^{(t)}, T^{K_t}, \mathrm{Sep}(K_i),$ and $\mathrm{Res}(K_i)$, Algorithm 4 will return $\mathcal{C}_G(K_i)$ and the set $L^{(i)}$ of super cliques of $T^{K_i}$.*

*Table 2.* Average running time (in seconds)

| $r$ | 0.34 | | 0.08 | |
|---|---|---|---|---|
| $|V|$ | $T_{\mathrm{ICP}}$ | $T_{\mathrm{CP}}$ | $T_{\mathrm{ICP}}$ | $T_{\mathrm{CP}}$ |
| 512 | 1.022 | 2.081 | 0.842 | 1.494 |
| 1024 | 8.048 | 15.579 | 5.586 | 9.974 |
| 2048 | 68.048 | 125.181 | 40.545 | 74.940 |
| 4096 | 707.484 | 1169.811 | 387.491 | 660.435 |

## 7. Experiment

We now evaluate our proposed SC-Trans algorithm. It is integrated into the Clique-Picking (CP) algorithm by replacing its Step 2 in Algorithm 1. The improved version is denoted as ICP. We compare the practical performance of our improved ICP and the state-of-the-art CP algorithm in a series of experiments. Both methods are implemented using Julia. All experiments are run on a laptop with AMD

*Table 3.* Comparison between ICP and CP (the runtime difference $T_{\text{CP}} - T_{\text{ICP}}$ and $T_{\text{CP},2} - T_{\text{ICP},2}$ are measured in seconds)

| $|V|$ | | 1024 | | | | 2048 | | |
|---|---|---|---|---|---|---|---|---|
| $r$ | $\theta$ | $T_{\text{CP}} - T_{\text{ICP}}$ | $T_{\text{CP},2} - T_{\text{ICP},2}$ | $T_{\text{CP},2}/T_{\text{ICP},2}$ | $\theta$ | $T_{\text{CP}} - T_{\text{ICP}}$ | $T_{\text{CP},2} - T_{\text{ICP},2}$ | $T_{\text{CP},2}/T_{\text{ICP},2}$ |
| 0.04 | 0.0381 | 2.17 | 2.33 | 5.34 | 0.0189 | 20.45 | 17.99 | 12.27 |
| 0.06 | 0.0243 | 2.84 | 3.30 | 7.56 | 0.0126 | 27.87 | 24.19 | 17.16 |
| 0.12 | 0.0108 | 5.01 | 4.73 | 18.43 | 0.0054 | 45.58 | 37.41 | 39.70 |
| 0.24 | 0.0044 | 7.02 | 7.35 | 47.59 | 0.0022 | 60.40 | 76.63 | 95.35 |
| 0.46 | 0.0015 | 8.42 | 9.20 | 153.55 | 0.0007 | 80.97 | 81.45 | 264.54 |

Ryzen9 2.7GHz and 16G RAM. All the implementations use only one thread of execution and the running time is measured for exact counting.

We use the minimal triangulation method to generate chordal graphs (Dethlefsen & Højsgaard, 2005). The replicates are generated by first generating a undirected graph of $|V|$ vertices and $\rho \cdot \binom{|V|}{2}$ edges, and then, the triangulation is made until the resulting graph is chordal. Graph density $r$ is measured by $|E|/|E_{\max}|$ with $|E_{\max}| = |V|(|V|-1)/2$. For each $|V|$, the parameter $\rho$ is adjusted to make the average number of edges of resulting chordal graphs equals to $r \cdot \binom{|V|}{2}$.

**Varying numbers of graph vertices.** We first tested the performance of ICP and CP in various vertices of chordal graphs. We performed experiments with $|V| = 512, 1024, 2048, 4096$. In each experiment we chose $r$ as either $r = 0.34$ or $r = 0.08$ and generated ten chordal graphs for each number of graph vertices. Let $T_{\text{CP}}$ denote the average running time of CP algorithm, and let $T_{\text{ICP}}$ denote the average running time of our ICP algorithm. The experimental results are shown in Table 2, Figure 3(a) and (b). Our ICP clearly consistently performs better and solve within less amount of time.

**Varying graph densities.** We then tested ICP and CP over various specification of graph density $r$. We performed experiments with $r = 0.04, 0.06, 0.12, 0.24, 0.46$. In each experiment we chose $|V|$ as either $|V| = 1024$ or $|V| = 2048$ and generated ten chordal graphs for each graph density. As shown in Table 3 and Figure 3(c), the difference in running time between the two algorithms becomes more pronounced with the increase of $r$. This trend is attributed to the fact that denser graphs generally correspond to a lower value of $\theta = m/(|V| + |E|)$, which enhances the performance advantage of our proposed method.

Recall that our proposed algorithm enhances Step 2 of Algorithm 1. When focusing solely on the computational cost of this step, the advantage of our method becomes even more evident. Let $T_{\text{CP},2}$ denote the average running time of Step 2 of CP algorithm, and let $T_{\text{ICP},2}$ denote the average running time of ICP for the same step. Table 3 presents both the difference and the ratio between these average running times. Figure 3(d) shows the average runtime difference of

Step 2 of CP and ICP. The results highlight substantial improvements achieved by our method in terms of efficiency.

# 8. Conclusion

In this work, we propose an enhancement to the Clique-Picking algorithm (Wienöbst et al., 2023) by avoiding the intensive and repeated generation of $\mathcal{C}_G(K_j)$ for each clique $K_j$ of a chordal graph $G$. Our improvement introduces a higher-level structure, termed a super clique, within the clique tree. We demonstrate that an efficient transfer of super cliques is possible between two clique trees with different choices of $K_j$ as the root. The proposed algorithm significantly reduces the computational cost of Step 2 in Algorithm 1.

# Acknowledgements

This work was supported in part by the National Natural Science Foundation of China under Grant No. 12431009; the National Key R&D Program of China under Grant No. 2020YFA0714102.

# Impact Statement

This paper presents work whose goal is to advance the field of machine learning and causal analysis. Our work facilitates more efficient causal discovery and decision-making processes, particularly in scenarios where high-dimensional data are involved. This advancement contributes to improving the feasibility of real-world applications such as automated causal inference in healthcare, policy-making, and artificial intelligence systems.

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

## A. Notation List

Table 4. Table of Frequently Used Notations

| Notation | | Meaning |
|---|---|---|
| $T^K$ | Rooted clique tree | Clique tree with rooted $K$ |
| $\pi(K)$ | | A permuted ordering of the vertices in $K$ |
| $G^{\pi(K)}$ | | The union of all DAGs within the MEC represented by $G$ that have topological orderings beginning with $\pi(K)$ |
| $G^K$ | | $\cup_\pi G^{\pi(K)}$ |
| $G[C]$ | Induced subgraph | The induced subgraph of $G$ on a vertex set $C$ |
| $\mathcal{C}_G(K)$ | | The set of undirected connected components of $G^K[V \setminus K]$ |
| $\mathrm{Sep}(K)$ | | The set of separators for $T^K$ |
| $\mathrm{Res}(K)$ | | The set of residuals for $T^K$ |
| $\mathrm{SK}_{p^+}^{(i)}$ | Super clique | The clique set contains clique header $K_p$ and the clique tails of $K_p$ within $T^{K_i}$ |
| $L^{(i)}$ | | The set of super cliques within $T^{K_i}$ |
| $\mathrm{SR}_{p^+}^{(i)}$ | Super residual | The set of the residuals for $T^{K_i}$ corresponding to the cliques within $\mathrm{SK}_{p^+}^{(i)}$ |

## B. D-Numbering

We now discuss D-numbering (Leimer, 1993; Guo & Wang, 2010), which offers an effective approach for analyzing clique sequence.

**Definition B.1.** (D-numbering) For a UCCG $G$, an order $\alpha$ of its vertices is called a D-numbering if there exists an RIP clique sequence $K_1, \ldots, K_m$ with corresponding residuals $R_1, \ldots, R_m$ such that

$$\alpha(R_1) = \{n, n-1, \ldots, n - |R_1| + 1\}, \ldots, \alpha(R_m) = \{|R_m|, \ldots, 2, 1\},$$

where $n = |V|$ is the number of vertices in $G = (V, E)$.

Note the above definition only specifies a set-to-set relation. Each $R_j$ is mapped to a set of numbers; however, the specific ordering of vertices within each $R_j$ is not defined. Therefore, generating a D-numbering from an RIP sequence does not yield a unique result, as multiple D-numberings can correspond to the same RIP sequence. For example, consider an RIP sequence for the graph in Figure 1(a): $K_1 = \{a, b, c\}$, $K_2 = \{b, c, d\}$, $K_3 = \{b, e\}$, $K_4 = \{e, f\}$, $K_5 = \{b, g, j\}$, $K_6 = \{b, g, i\}$ and $K_7 = \{b, h, j\}$. From this RIP sequence, we can determine the mapping $\alpha$ for the residuals:

$$\alpha(a, b, c) = \{10, 9, 8\}, \alpha(d) = \{7\}, \alpha(e) = \{6\}, \alpha(f) = \{5\}, \alpha(g, j) = \{4, 3\}, \alpha(i) = \{2\}, \alpha(h) = \{1\}.$$

Correspondingly, one of the D-numberings for the vertices can be defined as:

$$\alpha(b) = 10, \alpha(a) = 9, \alpha(c) = 8, \alpha(d) = 7, \alpha(e) = 6, \alpha(f) = 5, \alpha(j) = 4, \alpha(g) = 3, \alpha(i) = 2, \alpha(h) = 1.$$

Another D-numbering is

$$\alpha(a) = 10, \alpha(c) = 9, \alpha(b) = 8, \alpha(d) = 7, \alpha(e) = 6, \alpha(f) = 5, \alpha(j) = 4, \alpha(g) = 3, \alpha(i) = 2, \alpha(h) = 1,$$

which only changes the vertex order in residual $R_1$.

We can see that a D-numbering is also a perfect elimination ordering, meaning it represents a DAG of $G$.

**Theorem B.2.** (*Leimer, 1993*). *Any D-numbering must be a perfect elimination ordering.*

It is well known that any perfect elimination ordering can represent a DAG in a Markov equivalent class (Wienöbst et al., 2023). Therefore, for any D-numbering (which is a perfect elimination ordering by Theorem B.2), there exists a corresponding DAG in the MEC. On the other hand, a DAG in the MEC can be represented by one or more D-numbering(s). This statement correspond to Lemma B.3 below, respectively.

**Lemma B.3.** *Every DAG in the Markov equivalence class represented by $G$ can be represented by at least one D-numbering.*

*Proof.* Every DAG in a Markov equivalence class represented by $G$ can be represented by the ordering generated by MCS algorithm. The ordering of a UCCG obtained from the MCS algorithm is also a D-numbering (Blair & Peyton, 1993). □

## C. D-numbering, Clique Sequence and Root-Selected Essential Graph

Recall from Section 3 that, for a chordal graph $G$ with some selected maximal clique $K_1$, the root-selected essential graph $G^{K_1}$ is critical to the construction of the Clique-Picking algorithm of Wienöbst et al. (2023). We now discuss in more details how to relate D-numbering and clique sequence to this $G^{K_1}$.

In fact, with the concept of D-numbering, we can construct $G^{K_1}$ in the following way. First, for a chordal graph $G$, we find all RIP clique sequences beginning with $K_1$. Then, for each clique sequence, we enumerate all its D-numberings and construct the corresponding DAGs. The union of these DAGs is exactly $G^{K_1}$. This approach of constructing $G^{K_1}$ help us to identify the edges' direction and undirected connected component (UCCG) of $G^{K_1}$ from a different perspective. In this section, we address the edge direction determination. The UCCG identification for $G^{K_1}$ is left for the next section.

We start by considering a *specific* RIP sequence $\mathcal{S} = (K_1, \ldots, K_m)$. Suppose $x$ and $y$ are two adjacent vertices in $G$. We consider two cases below:

1. If $x$ and $y$ are in two different residuals $R_p$ and $R_q$, respectively with $p < q$. In this case, for any D-numbering $\alpha$ of $\mathcal{S}$, we have $\alpha(x) > \alpha(y)$. Recall a D-numbering is a perfect elimination ordering. This implies that we always have $x \to y$ in the DAGs represented by all D-numberings of $\mathcal{S}$.

2. If $x$ and $y$ are in an identical residuals $R_p$, then relative order between $\alpha(x)$ and $\alpha(y)$ can be arbitrary. The means the edge direction between $x$ and $y$ can be arbitrary among the DAGs represented by the D-numberings of $\mathcal{S}$.

Consider all the D-numberings of the RIP clique sequence $\mathcal{S}$, and denote $G^{\mathcal{S}}$ as the union of DAGs represented by these D-numberings. Then, we can conclude: (1) the endpoints of any directed edge in $G^{\mathcal{S}}$ are in different residuals and (2) the endpoints of any undirected edge in $G^{\mathcal{S}}$ are in the same residual.

Instead of a single RIP sequence, let us now consider all RIP sequences that start with $K_1$. For two adjacent vertices $x, y$ in $G$, it is possible that

(1) there exists an RIP sequence $\mathcal{S}_1 = (K_1, K_2, \ldots, K_m)$, such that $x$ and $y$ are in two different residuals $R_p$ and $R_q$, respectively with $p < q$. Then, we have $x \to y$ as an edge in $G^{\mathcal{S}_1}$;

(2) there exists a permuted RIP sequence $\mathcal{S}_2 = (K_{\sigma(1)}, K_{\sigma(2)}, \ldots, K_{\sigma(m)})$, such that $x$ and $y$ are in two different residuals $R_{\sigma(p')}$ and $R_{\sigma(q')}$, respectively but with $p' > q'$. Then, we have $x \leftarrow y$ as an edge in $G^{\mathcal{S}_2}$.

When both (1) and (2) in the above happen, the edge direction between $x$ and $y$ is undirected in $G^{K_1} = \cup_{\mathcal{S}} G^{\mathcal{S}}$, which is a union with respect to all RIP sequence $\mathcal{S}$ that starts with $K_1$.

In summary, the above discussion indicates we can construct $G^{K_1}$ by enumerating all clique sequences starting with $K_1$, enumerating all their D-numberings and construct the corresponding DAGs. Then, $G^{K_1}$ is simply the union of these DAGs. More importantly, if the relative order of two maximal cliques ($K_p$ and $K_q$) can vary across different RIP sequences, the direction of the edge between two vertices in their residuals may potentially be undirected in $G^{K_1}$.

## D. Super Cliques and Super Residuals from the Perspective of RIP Sequences

Now, we examine super clique, super residual and undirected connected component (UCCG) from the perspective of RIP clique sequence alone, without relying on the clique rooted trees. This provides more theoretical tools for establishing Theorem 5.3.

Recall from Section C, to construct some root-selected essential graph $G^{K_1}$, we have to consider all RIP sequence starting with $K_1$. Fortunately, this exhaustive process is not necessary. Instead, the key question is: how can we characterize $G^{K_1}$ using a single RIP sequence that starts with $K_1$?

As discussed in Section C, if the relative order of two maximal cliques ($K_p$ and $K_q$) can vary across different RIP sequences, the direction of the edge between their residuals may potentially remain undetermined. The following lemma implies that this uncertainty can be identified using a single clique sequence. Specifically, if their separators satisfy $S_p \subsetneq S_q$, then the relative order of $K_p$ and $K_q$ is interchangeable, and the edge between their residuals may remain undetermined.

**Lemma D.1.** *Let $K_1, K_2, \ldots, K_m$ be an RIP sequence of the clique set $\mathcal{K}_G$ in a UCCG $G$. If $S_p$ is a proper subset of $S_q$ for any $p, q = 2, \ldots, m$ with $p < q$, then there exists a permutation $\sigma$ such that $\sigma(1) = 1$ and $s > t(p = \sigma(s), q = \sigma(t))$ in the RIP sequence $K_{\sigma(1)}, K_{\sigma(2)}, \ldots, K_{\sigma(m)}$.*

*Proof.* We can assume without loss of generality $q = m$. The proof is by induction on $m$. The case $m = 2$ is trivial. Let $K_1, \ldots, K_m$ be an RIP sequence for some $m \geq 3$. There is a $p$ such that $S_p \subsetneq S_m$.

Case 1: $p = m - 1$.

Then we define $\sigma(m-1) = m$, $\sigma(m) = m - 1$, and $\sigma(k) = k, \forall k < m - 1$. $S_m \setminus S_{m-1}$ is not in $K_1, \ldots, K_{m-2}$, so there is a clique $K_r$ such that $K_m \cap (K_1 \cup \cdots \cup K_{m-2}) \subset K_r$. Meanwhile, $K_{m-1} \cap (K_1 \cup \cdots \cup K_{m-2} \cup K_m)$ must be a subset of $K_m$. Hence $K_{\sigma(1)}, K_{\sigma(2)}, \ldots, K_{\sigma(m)}$ is an RIP sequence in this case.

Case 2: $p < m - 1$.

Without loss of generality, assume that $S_p$ is a proper subset of $S_{m-1}$. Using the induction hypothesis there is a permutation $\sigma$ such that $K_{\sigma(1)}, K_{\sigma(2)}, \ldots, K_{\sigma(m-1)}$ is an RIP sequence. Then we get

$$K_{\sigma(1)}, \ldots, K_{\sigma(r-1)}, K_m, K_{\sigma(r)}, \ldots, K_{\sigma(m-1)}$$

where $\sigma(r) = m - 1$. For $\forall w > t, S_m \setminus S_p$ is not in the $K_m \cap K_{\sigma(w)}$. Meanwhile we have $S_p = K_m \cap (K_{\sigma(1)} \cup \cdots \cup K_{\sigma(t-1)})$ and $S_m = K_{\sigma(t)} \cap (K_{\sigma(1)} \cup \cdots \cup K_{\sigma(t-1)} \cup K_m)$. Hence the separators of the full sequences are also identical in this case. $\square$

For any two cliques $K_p$ and $K_q$ in the sequences $K_1, \ldots, K_m$, suppose their relative order can be reversed to get $K_{\sigma(1)}, K_{\sigma(2)}, \ldots, K_{\sigma(m)}$, where $p < q$ but $\sigma^{-1}(p) > \sigma^{-1}(q)$. This reflects the ambiguity in the direction of edges between $R_p$ and $R_q$. In fact, this uncertainty further indicates the residuals $R_p$ and $R_q$ can be potentially merged to form an undirected connected components in $\mathcal{C}_G(K_1)$.

For example, consider again the graph in Figure 1(a). The graph has the RIP sequence $K_1, K_2, K_3, K_4, K_5, K_6, K_7$. Because $S_5 \subsetneq S_6$, the graph also has the RIP sequence $K_1, K_2, K_3, K_4, K_6, K_5, K_7$, which is obtained by permuting $K_5$ and $K_6$. This indicates the uncertainty in the direction of the edge $g - i$ in $G^{K_1}$, which serves as a clue that $R_5$ and $R_6$ may form a UCCG in $G^{K_1}$.

However, checking interchangeability between two cliques is not sufficient to determine if we can merge their residuals. An additional requirement for merging their residuals is that $S_p$ does not separate $R_p$ and $R_q$ in $G$. Still consider our example in Figure 1. $S_3$ is a proper subset of $S_2$. But there is no need to merge $R_2$ and $R_3$ as $S_2$ separates $R_2$ and $R_3$. This leads to the following definitions. They correspond to the clique header and clique tail in the main text, but the definition here is described in terms of clique sequence.

**Definition D.2.** $K_1, K_2, \ldots, K_m$ is an RIP sequence of $\mathcal{K}_G$ in UCCG $G$. For any $p = 2, \ldots, m$, we say $K_p$ is a *sequence clique header* within this sequence if for any $q = 2, ..., p$ (1) $S_q$ is not proper subset of $S_p$ or (2) $S_p$ separates $R_p$ and $R_q$ in $G$.

**Definition D.3.** Let $K_p, 1 < p \leq m$, be a sequence clique header within given $K_1, K_2, \ldots, K_m$. For any $q = p, ..., m$, we say that $K_q$ is a *sequence clique tail* following $K_p$ if (1) $S_p$ is a proper subset of $S_q$ and (2) $S_p$ does not separate $R_p$ and $R_q$ in $G$.

In fact, when a clique tree is given, the above definitions are equivalent to those presented in the main text from the perspective of the clique tree structure. Similarly, super cliques and super residuals can be defined from the viewpoint of a clique sequence.

**Definition D.4.** (Sequence super clique, sequence super residual) Within an RIP sequence, suppose $K_p$ is a sequence clique header and $K_{p_1}, \ldots, K_{p_r}$ are all its sequence clique tails.

   i. The clique set $\{K_p, K_{p_1}, \ldots, K_{p_r}\}$ is called a *sequence super clique*.

ii. The set of the residuals $\{R_p, R_{p_1}, \ldots, R_{p_r}\}$ is called a *sequence super residual*.

**Theorem D.5.** *For a given RIP sequence $K_1, \cdots, K_m$, $\mathcal{C}_G(K_1)$ consists of the subgraphs of $G$ induced by the sequence super residuals within $K_1, \cdots, K_m$.*

*Proof.* For a sequence clique header $K_p$, assume there are $u$ separators, of which $S_p$ is a proper subset. We denote them as $S_{p_1}, \ldots, S_{p_u}$, where $1 < p_1 < \cdots < p_u \le m$. For convenience, assume that for any $q = p_1, \ldots, p_r$, where $r < u$, $S_p$ does not separate $R_p$ and $R_q$ in $G$. This is the opposite for any $q = p_{r+1}, \ldots, p_u$. For any $q = p_1, \ldots, p_u$, from Lemma D.1, we know that there exists an RIP sequence $K_{\sigma(1)}, K_{\sigma(2)}, \ldots, K_{\sigma(m)}$ such that $\sigma(1) = 1$ and $s > t(p = \sigma(s), q = \sigma(t))$. When $q = p_1$, for a node $x$ in $R_p \cap S_q$ and any $y$ in $R_q$, assume $x \sim y$ in $G$. The edge $x - y$ must be directed as $x \to y$ in the DAG represented by the D-numbering generated from $K_1, K_2, \ldots, K_m$. At this point $x$ and $y$ are in $R_{\sigma(t)}$. Therefore, the edge $x - y$ can be either $x \to y$ or $y \to x$ in the DAG represented by the D-numbering generated from $K_{\sigma(1)}, K_{\sigma(2)}, \ldots, K_{\sigma(m)}$. As a result, the edge $x - y$ will remain undirected in $G^{K_1}$.

Now we know that $G^{K_1}[R_p, R_{p_1}]$ is an undirected graph. Similarly, if exists an $v = 2, \ldots, u$ and $y \in R_{p_1} \cap S_{p_v}$ such that $y \sim z$ in $G$ for any $z \in R_{p_v}$, the edge $y - z$ will remain undirected in $G^{K_1}$. For any $q = p_{r+1}, \ldots, p_u$ we know that $S_p$ separates $R_p$ and $R_q$ in $G$, so such $v$ should lie in $2, \ldots, r$. Recursively, $G^{K_1}[R_p, R_{p_1}, \ldots, R_{p_r}]$ is an undirected graph. In Definition D.2, we confirm that $K_p$ cannot be a sequence clique tail following another clique. There are only $K_{p_1}, \ldots, K_{p_u}$ such that $K_p$ and $K_{p_v}$ can be sequentially replaced for any $v = 1, \ldots, u$. Furthermore, $K_{p_1}, \ldots, K_{p_r}$ are sequence clique tails following $K_p$ as defined in D.3. According to the above description, only $R_{p_1}, \ldots, R_{p_r}$ and $R_p$ together induce an undirected graph. $\square$

# E. Technical Proofs for the Main Paper

*Proof of Theorem 5.3.* First, assume that there exists a clique $K_q$, where $q < p$, such that $K_q$ is not an ancestral clique of the sequence clique header $K_p$. For such a $K_q$, we have $S_q \subsetneq S_p$, and $S_p$ does not separate $R_p$ and $R_q$ in $G$. Without loss of generality, assume that there exist nodes $x \in R_p$ and $y \in R_q$ such that $x \sim y$ in $G$. Then, the union of $S_q$ and $\{x, y\}$ will form a clique in $\mathcal{K}_G$. This must be a child clique of $K_p$ according to the MCS algorithm. From the clique-intersection property, we know that $y$ must appear in every clique along the unique path from $K_p$ to $K_q$. However, $y$ is not in $K_p$, leading to a contradiction.

The same argument applies for the second case. There cannot exist a non-descendant clique $K_q$ of $K_p$ such that $S_p \subsetneq S_q$, and $S_p$ does not separate $R_p$ and $R_q$ in $G$. $\square$

*Proof of Propositions 6.2 & 6.3.* Denote $K_1 \cap \cdots \cap K_{p-1}$ as $H_p$. For $p = i+1, \ldots, m$, the proposition holds naturally, as the $H_{p-1}$ remains unchanged after the permutation. For $p = 2, \ldots, i-1$, we have $K_p = K_{\sigma(p+1)}$. In $K_1, \ldots, K_m$, denote the parent clique of $K_p$ as $K_q$. Thus, $K_p \cap H_{p-1}$ lies within $K_q$. Lemma 6.1 implies that $H_{\sigma(p)} = H_{p-1} \cup K_i$, which means $S_{\sigma(p+1)} =$

$$K_{\sigma(p+1)} \cap H_{\sigma(p)} = (K_{\sigma(p+1)} \cap H_{p-1}) \cup (K_{\sigma(p+1)} \cap K_i).$$

The first item is equal to $K_p \cap H_{p-1}$, which is contained within $K_q$. The second item is a subset of $K_i \cup H_{i-1}$, which in turn is contained within $K_1$. It follows naturally that $S_{\sigma(p+1)}$ remains contained within $K_q$ when $K_p \cap K_i = \emptyset$. When $K_p \cap K_i \ne \emptyset$, we also have $K_p \cap K_1 \ne \emptyset$, indicating that $K_1$ must be the parent of $K_p$, i.e., $t = 1$. In this case, $S_{\sigma(p+1)}$ is still contained within $K_q$. $\square$

*Proof of Theorem 6.4.* First, $K_i$ cannot be a clique header within $T^{K_i}$. Thus, for $\mathrm{SK}_{i+}^{(t)}$ of $T^{K_t}$, it will break down into several super cliques with respect to $T^{K_i}$. Any child clique of $K_i$ must be a clique header within $T^{K_i}$. From Propositions 6.1 & 6.3, the separator $S_p$ remains the same in $\mathrm{Sep}(K_i)$ and $\mathrm{Sep}(K_t)$ for any $p \in [m] \setminus \{i, t\}$. For any child clique of $K_i$, denote it temporarily as $K_q$. All descendant cliques of $K_q$ in $\mathrm{SK}_{i+}^{(t)}$ will be clique tails that follows clique header $K_q$ within $T^{K_i}$.

Second, $K_t$ must be a clique header within $T^{K_i}$. The child clique of $K_t$ must be a clique header within $T^{K_t}$. However, the child clique of $K_t$ may also be a clique tail that follows clique header $K_t$ within $T^{K_i}$. For any child clique of $K_t$, denote it temporarily as $K_p$. From Definition 5.1, if $S_t$ in $\mathrm{Sep}(K_i)$ is a proper subset of $S_p$ in $\mathrm{Sep}(K_i)$, then $K_p$ will be a clique tail that follows clique header $K_t$ within $T^{K_i}$. Otherwise, $K_p$ remains a clique header within $T^{K_i}$.

Thus, aside from the two points mentioned above, the clique header within $T^{K_t}$ remains the same within $T^{K_i}$. The separator $S_p$ is identical in both $\mathrm{Sep}(K_i)$ and $\mathrm{Sep}(K_t)$ for any $p \in [m] \setminus \{i, t\}$. Therefore, the set of clique tails that follows a clique header within $T^{K_t}$ is the same as that within $T^{K_i}$.

Proposition 6.3 tells us that for any $p \in [m] \setminus \{i, t\}$, $R_p$ is the same in both $\mathrm{Res}(K_i)$ and $\mathrm{Res}(K_t)$. Additionally, it is clear that $R_t = K_t \setminus (K_i \cap K_t)$ in $\mathrm{Res}(K_i)$. The super cliques described in SC-Trans-Op($\cdot$)(lines 3-8) do not involve $K_t$. Similarly, any super clique described in Trans($\cdot$)(lines 11-14) also does not involve $K_t$. However, for the super cliques described in Trans($\cdot$)(line 9-10), $K_t$ is involved. Thus, $\mathrm{SR}_{t+}^{(i)}$ of $T^{K_i}$ is the union of $R_t = K_t \setminus (K_i \cap K_t)$ in $\mathrm{Res}(K_i)$, and the super residuals of the super cliques with respect to $T^{K_t}$ proposed in SC-Trans-Op($\cdot$)(line 10). $\qquad\square$

*Proof of Theorem 4.1.* We begin by showing that any clique header within $T^{K_1}$ and the following clique tails are obtained from $\mathrm{Sep}(K_1)$. The conditions for a clique header and a clique tail are opposites. For the sequence $p = 2, \ldots, m$, we proceed with the following procedure: If there is an ancestral clique $K_q$ of $K_p$ such that $S_q \subsetneq S_p$ (with $S_p$ and $S_q$ being in $\mathrm{Sep}(K_1)$), then $K_p$ will be a clique tail following $K_q$. Otherwise, $K_p$ will be a clique header within $T^{K_1}$. For any super clique $\mathrm{SK}_{p+}^{(1)}, 2 \le p \le m$, of $T^{K_1}$, the clique $K_p$ will be identified in the $p$-th step, and all other cliques in $\mathrm{SK}_{p+}^{(1)}$, i.e., the clique tails following $K_p$, denoted as $K_{p_1}, \ldots, K_{p_r}$, will be identified in the $p_1, \ldots, p_r$-th steps.

For any $i = 2, \ldots, m$, in $T^{K_t}$, $K_t \to K_i$, and by reversing $K_t \to K_i$ to $K_i \to K_t$, the resulting tree is $T^{K_i}$, where $K_t$ is the parent of $K_i$ in $T^{K_1}$. There are $m$ separators $S_1, \ldots, S_m$ in $\mathrm{Sep}(K_t)$. Let $S_t = K_t \cap K_i$ and $S_i = \emptyset$. Now $S_1, \ldots, S_m$ will form $\mathrm{Sep}(K_i)$. Similarly, there are $m$ residuals $R_1, \ldots, R_m$ in $\mathrm{Res}(K_t)$. Let $R_t = K_t \setminus (K_t \cap K_i)$ and $R_i = K_i$. Now $R_1, \ldots, R_m$ will form $\mathrm{Res}(K_i)$. Next proof is by induction on $i$. Using the induction hypothesis we have computed the set of super cliques and super residuals with respect to $T^{K_t}(t < i)$. Thus we will derive the set of super cliques and super residuals with respect to $T^{K_i}$ from these with respect to $T^{K_t}$ by Theorems 6.4. $\qquad\square$

*Proof of Theorem 4.2.* To compute $\mathcal{C}_G(K_1)$, we need to implement the procedure in order $p = 2, \ldots, m$, as proposed in the proof of Theorem 4.1. For each clique $K_p$, there are most $p - 1$ ancestral cliques. So we obtain the following bound:

$$\sum_{i=2}^{m} (i - 1) = \frac{1}{2}(m^2 - m).$$

For $i = 2, \ldots, m$, we can easily get $\mathrm{Sep}(K_i)$ and $\mathrm{Res}(K_i)$ from $\mathrm{Sep}(K_t)$ and $\mathrm{Res}(K_t)$, where $K_t$ is the parent clique of $K_i$ in $T^{K_1}$. Function SC-Trans-Op($\cdot$) describes the difference between the set of super cliques with respect to $T^{K_i}$ and $T^{K_t}$. SC-Trans-Op($\cdot$)(lines 3-8) requires identifying all child cliques of $K_i$ in $T^{K_t}$, while SC-Trans-Op($\cdot$)(lines 9-10) requires identifying all child cliques of $K_t$ in $T^{K_t}$. The sum of the number of child cliques of $K_t$ and $K_i$ is bounded in $m - 2$. In the worst case, for any $K_i, i = 2, \ldots, m$, the computation of $\mathcal{C}_G(K_i)$ is bounded by $m$. Therefore, the total computation for $\mathcal{C}_G(K_2), ..., \mathcal{C}_G(K_m)$ is bounded by $m(m - 1)$. In conclusion, SC-Trans algorithm has a time complexity of $\mathcal{O}(m^2)$. $\qquad\square$

# F. Working Example

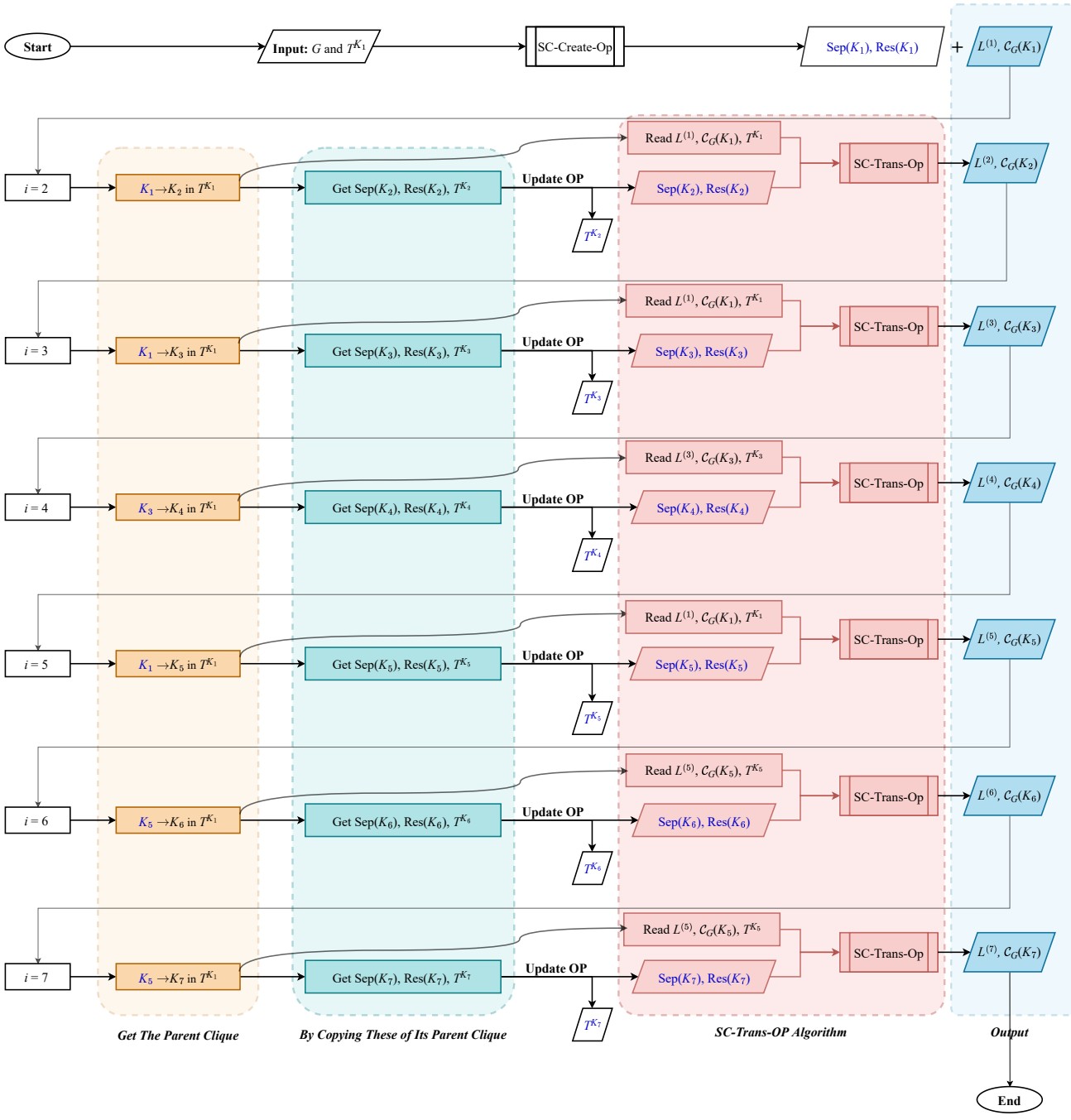

*Figure 4.* A working example of our SC-Trans algorithm (**Input:** UCCG $G$ in Figure 1(a) and rooted clique tree $T^{K_1}$ in Figure 1(b), **Output:** $\mathcal{C}_G(K_1), \ldots, \mathcal{C}_G(K_7)$). The detail of Update OP is presented in Algorithm 2 Lines 5-6 & 8.

