# OpenReview forum: "An Improved Clique-Picking Algorithm for Counting Markov Equivalent DAGs via Super Cliques Transfer"
_ICML.cc/2025/Conference — ICML 2025 oral_

### Official Review · Reviewer_W4jQ · 2025-02-19

**Overall Recommendation:** 4

**Summary:**

The paper proposed a more efficient way to improve the existing work by Wienobst et al on counting the number of DAGs in an MEC. The authors argue that the previous approach suffers in the case where there are multiple maximal cliques in a MEC. The proposed improvement is to make use of structure overlaps between these maximal cliques. To facilitate the construction of the method, the paper introduces two key notions named super clique and super residual. The key contribution in this paper is to come up with an efficient transfer operation of super cliques from one maximal clique-directed tree to another to avoid counting redundant structures.

## update after rebuttal

I have not changed my score because I support this paper to be accepted.  I think the paper has made a non-trivial contribution to improving the efficiency of the clique-picking algorithm by Wienobst et al. The efficiency is significant for scalable causal discovery.

**Claims And Evidence:**

Mostly yes with some minor issues I raised in the questions.

**Essential References Not Discussed:**

No

**Experimental Designs Or Analyses:**

I have checked the experiment presented in the paper. I don't see any issue.

**Methods And Evaluation Criteria:**

Mostly yes. I think the experiment should try to push for very large and dense graphs in order to show the merits.

**Other Comments Or Suggestions:**

- "Notably, Wien¨obst et al. (2023) introduce the Clique- Picking (CP) algorithm, which is a polynomial-time algorithm for counting MECs.” I think it will be better to just say determining the size of an MEC or counting the number of DAGs in an MEC instead of saying counting MECs.
- The big O notations are inconsistent e.g. line 193,194 vs line 56
- It would be better to add an example in the paper to walk people through the algorithm to show how the algorithm works in one clean way.
- I suggest the authors to put Algorithm 2 instead of "SC-Trans algorithm” in Theorem 4.1 and Theorem 4.2 so that the authors don’t need to explicitly say: "For our SC-Trans algorithm in Algorithm 2, we have the following results.”
- The "Case 1, Case 2, Case 3” in section 6.2 can be wrapped by a number list.
- Line 427: "Algorithm1" -> "Algorithm 1"

**Other Strengths And Weaknesses:**

I think this work will have the most positive impact on large and dense graphs. This aligns with the original motivation why one even needs to have a polynomial-time counting algorithm. The non-trivial part of this paper is to come up with the notion of super cliques and residuals and innovatively incorporate them in an efficient transfer operation. I personally spent quite a lot of time to understand the idea and tried to come up with an alternative method, but I failed.

For weaknesses, there are a few typos, but it's very minor. I think the presentation of the paper can be improved by providing a working example in the appendix since there are various different graph concepts and it's hard to remember all the notations until I have read it many times back and forth. The most obvious weakness is the efficiency gain over the previous work. It does not save more than half of the time even for the size of 4096. When r is increased, the patterns in the left and right figures in Figure 3 are almost the same. It would be better to try more extreme cases to show different behaviors of the algorithm.

**Questions For Authors:**

1. In line 147, why G[e] and G[f] are two separate undirected connected components but not G[e,f] in C_{G}(K_{1})?

2. In line 201, how can C_{G}(K_{3}) contain G[b,c,d] given that K_{3} = {b,e} and C_{G}(K_{3}) is defined as the undirected connected components of G^{K}[V \ K], which is a subgraph that excludes K by definition? Do you mean G[a,c,d]?

3. Can I replace "super residuals” in Theorem 4.5 with just residuals except for the residuals from the root?

**Relation To Broader Scientific Literature:**

This work further improves on the previous polynomial-time counting algorithm by Wienobst et al. (2023). It is built on the previous approach with an improvement to avoid counting the number of DAGs in the overlapping members of the undirected connected components. The algorithm remains polynomial-time with respect to the number of maximal cliques.

**Theoretical Claims:**

Yes. I have checked the proofs in section D. I don't find any issue.

---

> ### Author Rebuttal · Authors · 2025-03-31
>
> Thank you for your careful reading and valuable feedback. We address  the comments  in details below.
>
> >**Weaknesses:***"...the paper can be improved by providing a working example in the appendix..."*
>
> Thanks for the suggestion. We plan to include a table summarizing all key notations and concepts, along with a detailed example illustrating how the algorithm works. These additions, to be placed in the appendix, will enhance the clarity and accessibility of our method for a broader audience.
>
> >*"The most obvious weakness is the efficiency gain over the previous work .... try more extreme cases to show different behaviors of the algorithm."*
>
> Thank you for this insightful comment. We agree that evaluating performance on larger and denser graphs is important for highlighting the efficiency gains. The state-of-the-art Clique-Picking (CP), proposed by Wienöbst et al. (2023) and presented as Algorithm 1 in our paper, is already quite efficient. Our work focuses on improving Step 2 of Algorithm 1 through our proposed algorithm ( ICP).
>
> For Step 2, the computational complexity of  ICP and CP is $\mathcal{O}(m^2)$ and $\mathcal{O}(m(|V| + |E|))$, respectively. This represents a strict and substantial improvement when $m(>1)$ is of moderate size and $|V|+|E|$ is large.
>
> To provide additional evidence, we conducted experiments across various levels of graph edge density $r=|E|/|E_{\max}|$, where $|E_{\max}|=|V|(|V|-1)/2$. We performed experiments for $|V|=1024$ and $2048$, and the results are reported in the tables below. As the edge density $r$ increases, the difference in running time between the two algorithms becomes more pronounced. This trend is attributed to the fact that denser graphs generally correspond to a lower value of $\theta=m/(|V|+|E|)$, which enhances the performance advantage of our proposed method.
>
> **Table1: Comparison between ICP and CP**
> |||||||
> |:-:|:-:|:-:|:-:|:-:|:-:|
> ||| $\vert V\vert =1024$ ||||
> |$r$|0.06|0.09|0.15|0.27|0.43|
> |$\mathrm{CP}-\mathrm{ICP}$ (seconds)|3.752|4.40|5.71|7.10|8.73|
> |||$ \vert V\vert=2048$||||
> |$r$|0.04|0.06|0.10|0.22|0.49|
> |$\mathrm{CP}-\mathrm{ICP}$ (seconds) |19.45 |34.47|41.54|59.24|62.24|
>
> When isolating the comparison for Step 2 of Algorithm 1, the advantage of our algorithm becomes even prominent.  Denote $T_{\mathrm{CP},2}$ as the average running time of Step 2  executed as in Wienöbst et al. (2023), denote $T_{\mathrm{ICP},2}$ as the average running time of our approach for Step 2. The table below presents both their difference and ratio. The results highlight substantial improvements achieved by our method in terms of efficiency.
>
> **Table2: Comparison between Step 2 of ICP and CP**
> |||||||
> |:-:|:-:|:-:|:-:|:-:|:-:|
> ||| $ \vert V\vert=1024$ ||||
> |$r$|0.06|0.09|0.15|0.27|0.43|
> | $T_{\mathrm{CP},2}-T_{\mathrm{ICP},2}$ (seconds) | 3.31 | 4.65 |5.66 |8.04| 9.28 |
> |$T_{\mathrm{CP},2} / T_{\mathrm{ICP},2}$ |7.88|19.02|26.80|65.91|159.57|
> ||| $ \vert V\vert=2048$ ||||
> |$r$|0.04|0.06|0.10|0.22|0.49|
> |$T_{\mathrm{CP},2}-T_{\mathrm{ICP},2}$ (seconds) |17.99| 24.19|37.41 |72.44 |81.45 |
> |$T_{\mathrm{CP},2} / T_{\mathrm{ICP},2}$ |10.05|16.42|29.71|92.31|318.54|
>
> The full results will be included as new figures and tables. We believe these additional experiments significantly enhance the evidence to support the advantages of the proposed method.
>
> > **Other Comments Or Suggestions:**
>
> Thank you for these detailed suggestions. We will carefully revise the manuscript to address each of these points in the final version.
>
> > **Questions For Authors:**
>
> > **Q1**
>
> Thank you for the question. The orientation of the UCCG  $G$ cannot form new  $v$-structures or directed cycles. When  $K_1$ is selected the root, the edge between vertices $b$ and $e$ will be oriented as $b \rightarrow e$. Suppose $f \rightarrow e$, this would lead to  $v$-structure: $b\rightarrow e\leftarrow f$. Therefore, to avoid this, it must be  $e\rightarrow f$ in $G^{K_1}$, and $G[e]$ and $G[f]$ remain two separate undirected connected components.
> >**Q2**
>
> You are right. We deeply apologize for this typo. $\mathcal{C}_G(K_3)$ should contain $G[a,c,d]$, *not* $G[b,c,d]$. We will carefully proofread the paper and correct the typos in the final version.
>
> >**Q3**
>
> We assume your question refers to Theorem 5.3. The answer is no—replacing "super residuals" with individual residuals would lead to incorrect identification of undirected connected component, and hence incorrect counts of Markov Equivalent DAGs via Equation (2).
>
> Take the example in Figure 1: the subgraph $G[g,h,i,j]$ forms an undirected connected component in $G^{K_1}$. This component arises from merging the residuals $R_5$={ $g,j$ }, $R_6$={ $i$ } and $R_7$={ $h$ } into a single super residual. However, these residuals on their own—$R_5,R_6,R_7$—do not, by definition, form undirected connected components. Therefore, super residuals are essential for correctly identifying related structures and correctly counting Markov Equivalent DAGs.

---

> > ### Comment · Reviewer_W4jQ · 2025-04-03
> >
> > I thank the authors for the response. I appreciate the new experiments posted above. Please include it in the camera-ready version.

---

### Official Review · Reviewer_ev8s · 2025-03-09

**Overall Recommendation:** 4

**Summary:**

This paper proposes an improvement to the clique-picking algorithm introduced by Wienöbst et al. (2023) for counting Markov Equivalent Directed Acyclic Graphs (DAGs). The authors introduce super cliques and super residuals to reduce computational complexity when identifying undirected connected components (UCCGs) in a Completed Partially Directed Acyclic Graph (CPDAG). The proposed Super Cliques Transfer Algorithm optimizes the recursive clique-selection process, leading to a computational cost reduction from $O(m(|V|+|E|)$ to $O(m^2)$ where $m$ is the number of maximal cliques. Experiments on randomly generated chordal graphs demonstrate that the improved algorithm (ICP) outperforms the original Clique-Picking (CP) algorithm in runtime.

**Claims And Evidence:**

- The proposed Super Cliques Transfer Algorithm reduces the computational complexity of counting MEC sizes from $O(m(|V|+|E|)$ to $O(m^2)$ is supported by the theoretical analysis in Theorems 4.1 and 4.2.
- The claim of the improved algorithm speeds up MEC size counting is supported by the experimental results in Table 2 and Figure 3.

**Essential References Not Discussed:**

Not sure. But the literature seems to be well cited in the discussion in paragraph 3-4 in the introduction.

**Experimental Designs Or Analyses:**

The experiments are conducted to demonstrate the improvement in computational complexity. They all make sense to me.

**Methods And Evaluation Criteria:**

The paper follows a solid algorithmic and theoretical approach. The evaluation is conducted via simulation experiments.

**Other Comments Or Suggestions:**

- More discussion on the quantitive comparison between the computational complexity of proposed algorithm with the existing work will be appreciated. Currently, the only comparison is $O(m^2)$ v.s. $O(m(|V|+|E|)$. Is this a strict improvement? When is there a significant gap? Is there any special class of DAGs where the proposed algorithm is much better?
- It would be more illustrative to plot the experiment results in log-log plot.

**Other Strengths And Weaknesses:**

**Strength**:
- Reducing redundancy in clique-picking through transfer operations is well-motivated.
- The running examples provided in Figure 1 are very helpful for illustrating the mathematical concepts and understanding the algorithms.

**Weakness**:
- See suggestions and questions

**Questions For Authors:**

- How do we compare the proposed algorithm to other counting algorithms listed in the literature review -- thrid paramgraphs in the introduction -- in terms of both theoretical analysis and experimental comparison?

**Relation To Broader Scientific Literature:**

Bayesian network is widely used for scientific exploratory analysis, which is typically only identifiable up to so-called Markov equivalence class (MEC). To count the number of possible elements inside a given MEC is crucial for many downstream task and experimental degisn.

**Theoretical Claims:**

The paper presents several new theoretical constructs, including super cliques and super residuals, and proves their validity. Theorems 4.1 and 4.2 ensure the correctness and complexity of the Super Cliques Transfer Algorithm. While not carefully checked, the proofs appear well-structured and logically sound.

---

> ### Author Rebuttal · Authors · 2025-03-31
>
> Thank you for the valuable comments and remarks. We will address your questions and suggestions below.
>
> > **Other Comments Or Suggestions:**
> *"More discussion on the quantitive comparison between the computational complexity of proposed algorithm with the existing work will be appreciated."*
>
> Thanks for the comments. We have conducted additional experiments and will expand the related discussion as follows.
>
> In particular, we have  tested the proposed algorithm (ICP) and the previous algorithm (CP) over various specification of **graph edge density $r$**, which is measured by $|E|/|E_{\max}|$ with
> $|E_{\max}| = |V|(|V|-1)/2$. We performed experiments for $|V| = 1024$ and $2048$, and the results are summarized in the tables below. As the edge density $r$ increases, the difference in running time (measured in seconds) between the two algorithms becomes more pronounced. This trend is attributed to the fact that denser graphs generally correspond to a lower value of $\theta = m / (|V| + |E|)$, which enhances the performance advantage of our proposed method.
>
> **Table1: Comparison between ICP and CP**
> |||||||
> |:-:|:-:|:-:|:-:|:-:|:-:|
> ||| \|$V$\|=1024 ||||
> |$r$|0.06|0.09|0.15|0.27|0.43|
> |$\mathrm{CP}-\mathrm{ICP}$ (seconds) |3.75|4.40|5.72|7.10|8.73|
> ||| \|$V$\|=2048 ||||
> |$r$|0.04|0.06|0.10|0.22|0.49|
> |$\mathrm{CP}-\mathrm{ICP}$ (seconds) |19.45 |34.47|41.54|59.24|62.24|
>
> Recall that our proposed algorithm enhances Step 2 of Algorithm 1. When focusing solely on the computational cost of this step, the advantage of our method becomes even more evident. Let $T_{\mathrm{CP},2}$ denote the average running time of Step 2 as implemented in Wienöbst et al. (2023), and let $T_{\mathrm{ICP},2}$ denote the average running time of our improved approach for the same step. The table below presents both the difference and the ratio between these average running times. The results highlight substantial improvements achieved by our method in terms of efficiency.
>
> **Table2: Comparison between Step 2 of ICP and CP**
> |||||||
> |:-:|:-:|:-:|:-:|:-:|:-:|
> ||| $\vert V\vert=1024$ ||||
> |$r$|0.06|0.09|0.15|0.27|0.43|
> | $T_{\mathrm{CP},2}-T_{\mathrm{ICP},2}$ (seconds) | 3.31 | 4.65 |5.66 |8.04| 9.28 |
> |$T_{\mathrm{CP},2} / T_{\mathrm{ICP},2}$ |7.88|19.02|26.80|65.91|159.57|
> ||| $ \vert V\vert=2048$ ||||
> |$r$|0.04|0.06|0.10|0.22|0.49|
> |$T_{\mathrm{CP},2}-T_{\mathrm{ICP},2}$ (seconds) |17.99| 24.19|37.41 |72.44 |81.45 |
> |$T_{\mathrm{CP},2} / T_{\mathrm{ICP},2}$ |10.05|16.42|29.71|92.31|318.54|
>
> The full results will be included as new figures and tables in the final version of the manuscript. We believe these additional experiments significantly enhance the evidence to support the advantages of the proposed method.
>
> > **Other Comments Or Suggestions:**
> *"Is this a strict improvement? When is there a significant gap? Is there any special class of DAGs where the proposed algorithm is much better?"*
>
> In this work, we propose SC-Trans algorithm to improve Step 2 of Algorithm 1, which is the Clique-Picking (CP) algorithm by Wienöbst et al.(2023).  The computational complexity of our proposed algorithm is $\mathcal{O}(m^2)$, compared to $\mathcal{O}(m(|V| + |E|))$ for the corresponding step in the CP algorithm. This constitutes a strict and significant improvement when $m (> 1)$ is a moderate value and the graph size, i.e., $|V| + |E|$, is large. The improvement is especially pronounced for large graph where the ratio $m/(|V| + |E|)$ is small. This is supported by the numerical evidence as in our response to your previous comment (please refer to the tables above).
>
> > **Other Comments Or Suggestions:**
> *"It would be more illustrative to plot the experiment results in log-log plot."*
>
> Thanks for the suggestions. We will adjust the plot to better demonstrate the advantages of the proposed algorithm in the final version of the manuscript.
>
> > **Questions For Authors:**
> *"How do we compare the proposed algorithm to other counting algorithms"*
>
> Thank you for the question. The Clique-Picking (CP) algorithm proposed by Wienöbst et al. (2023) is the first known algorithm with polynomial complexity for this task and currently represents the state of the art. In their work, the CP algorithm has been shown to significantly outperform previous counting algorithms. Our proposed approach builds upon and improves the Clique-Picking algorithm, and therefore also outperforms existing alternatives.

---

> > ### Comment · Reviewer_ev8s · 2025-04-03
> >
> > I thank the authors for their helpful resposne. It addresses most of my concern. I will keep the score for acceptance.

---

### Official Review · Reviewer_y8Y8 · 2025-03-14

**Overall Recommendation:** 4

**Summary:**

This submission presents an improvement of the recent polynomial time algorithm for counting moral acyclic orientations of chordal graphs, a problem which lies at the core of counting Markov equivalent DAGs. The main idea of that algorithm compared to older iterative root-picking algorithms lay in picking root-cliques. However, the prior algorithm recomputed equivalent information multiple times when choosing a different sequence of root-cliques. The present submission can identify and avoid this behavior by using what the submission refers to as super clique structure. Overall this improves the efficiency of the general approach at the point where the next lookup assuming the specific choice of the root clique. Beside proving the improved asymptotic complexity bound, the submission includes small test indicating the increased efficiency also empirically.

**Claims And Evidence:**

No concerns.

**Essential References Not Discussed:**

None that I am aware of.

**Experimental Designs Or Analyses:**

I did not as the experiments are not the main contribution and the mild claims made about them seem unproblematic.

**Methods And Evaluation Criteria:**

The methods make sense for the claims being made.

**Other Comments Or Suggestions:**

None.

**Other Strengths And Weaknesses:**

The topic and contribution are relevant to the audience at ICML. The presentation is also satisfactory. In terms of content, I find the key ideas behind the improvement elegant and natural. I recommend acceptance of this submission.

**Questions For Authors:**

None.

**Relation To Broader Scientific Literature:**

The submission continues a line of research improving the runtime of algorithms for counting MAOs. This is also clearly described in the introduction of the submission.

**Theoretical Claims:**

I read only what is not in the appendix.

---

> ### Author Rebuttal · Authors · 2025-03-31
>
> We greatly thank you for the positive feedback and valuable recognition of our work's core ideas and contributions.

---

### Official Review · Reviewer_WqwP · 2025-03-15

**Overall Recommendation:** 4

**Summary:**

Paper addresses the computational complexity of finding the size of the so-called Markov equivalence class (MEC) encoding conditional dependencies. Conditional dependence properties are captured by d-separation property of the DAGs and the method counts DAGs in the MEC class (representing the same/equivalent causla relationships). Generally, size of MEC grows exponentially in number of vertices and efficient methods are needed.

The paper improves on the recent Clique Picking (CP) method of Wienobst et al. (2023) presented in Algorithm 1. in the paper. Authors propose novel click structures called Superclicks (Definition 5.2) and propose a novel Super Clique Transfer procedure to improve computational cost of generating undirected connected components $C_G(K)$ of the tree with selected root clique $K$, the step 2. of the Algorithm 1. Using Super Cliques (SCs) enable an efficient reuse of the structures across different rooted trees avoiding repetitive construction. As a result, after definitions and theoretical framework derivations (Theorem 4.1, 4.2) and further detailed in Sections 5&6, the complexity of step 2 of Algorithm 1 is reduced from $O(m(|V|+|E|))$ to $O(m^2)$.

This constitutes the main contribution of the paper, supported by numerical experiments on random choprdal graphs in Section 7. This is claimed to enhance a feasibility of real-world applications such as casual inference in healthcare and AI.

## update after rebuttal: Thanks to authors for updates, new experiments and clarifications regarding proofs. I found their arguments and experimental results convincing and raise my score to 4.

**Claims And Evidence:**

Overall, problem to solve is well motivated and laid out. Arguments are clear and readable, yet exposition is quite technical. Figure 1. helps a lot, but further visual helpers may improve appeal of the paper to wider community. Experimental evidence is rather frugal but conceptually sufficient I believe. Clarity of presentation of the evidence in Section 7, is on the other hand, recommended for a remake. See comments and Weaknesses 1 and 2 in respective sections.

**Essential References Not Discussed:**

NA

**Experimental Designs Or Analyses:**

Section 7 presents experiments (Figure 3, Tale 2) on rangom graphs with increasing number of vertices for two ratios of dependencies in them. Namely $r=0.08$ vs. $r=0.34$, where $r=m/|V|+|E|$ suitably chopsen to demonstrate strong side of the proposed SC-Trans algortihm whose efficiency should excel in settings, where number of DAGs $m$ is strongly smaller than number of vertices and edges.

While text in Section 7. claims the Table 2 and Fig.3 demonstrates this well, it is in fact quite hard to see from the experiments how the compute time scales with $r$.

$\textbf{Weakness 2}$: While I agree with experiments supporting the claim, it is just quite hard to read it aout from absolute numbers presented. It is suggested to use relative measure, e.g., a (run time diference ICP-CP) over changing $r$ or vertices, or other means to present the evidence of the main claim of the paper more clearly.

**Methods And Evaluation Criteria:**

Paper argument is mainly theoretical so the selected experiments on random graphs are rather verification and supportive evidence. In this sense they are sufficient. For details see respective sections below.

**Other Comments Or Suggestions:**

-typos, L076 ("product" instead of multiplication), L186 ("be be")

**Other Strengths And Weaknesses:**

Strengths: novelty, contribution, addressing a timely problem, theoretical arguments are well written (yet still lacking some rigour that is suggested to be added, see W1)

Other Minor Weaknesses: Applications and possible problems to be solved by proposed method

**Questions For Authors:**

See above

**Relation To Broader Scientific Literature:**

Introduction contains relevant mentions of the literature of the field with applications in epidemiology, biology ad economics with references mentioned.

**Theoretical Claims:**

As for Theoretical evidence, Theorems 4.1, 4.2, Lemma 6.1, Propositions 6.2 and 6.3. are clearly stated but it is unclear where the proof can be found. Supplementary Material contains only Jupyter Notebook implementation of the SC+Trans algorithm.

The sketch of the arguments is to be found in related sections for Lemma and Propositions, but some details are missing, e.g., 'easy to see' under Proposition 6.3, etc.

$\textbf{Weakness 1}$: Overall I find arguments convincing but theoretical argument would benefit from more detailed proofs in Supplementary Material. This would make the paper more complete and self-contained rendering it more useful to wider research community, I believe.

---

> ### Author Rebuttal · Authors · 2025-03-31
>
> We sincerely appreciate your feedback. We are excited that you have found our approach useful and novel. Please find below our response to your concerns.
>
> > **Claims And Evidence:** *"...further visual helpers may improve appeal of the paper to wider community..."*
>
> Thank you for the helpful suggestion. We will include a table summarizing all key notations and concepts, along with a detailed example illustrating how the algorithm works. These additions, to be placed in the appendix, will enhance the clarity and accessibility of our method for a broader audience.
>
> > **Theoretical Claims:** *"...but some details are missing, e.g., 'easy to see' under Proposition 6.3, etc."*
>
> Thank you for pointing this out. We will provide more detailed explanation texts  to improve clarity and rigor.  In particular, for *'easy to see' under Proposition 6.3*, we will add an explanation: "The edge $K_1\rightarrow K_i$ in $T^{K_1}$ will be redirected as $K_i\rightarrow K_1$ in $T^{K_i}$. This implies $K_1$ becomes a child clique of $K_i$ in $T^{K_i}$. As the root $K_i$ is the only ancestral clique of $K_1$ in $T^{K_i}$,  $K_1$ must be a clique header within $T^{K_i}$ by Definition 5.1."
>
>
> > **Weakness 1** *"...it is unclear where the proof can be found... Overall I find arguments convincing but theoretical argument would benefit from more detailed proofs..."*
>
> Thank you for your  feedback. We would like to clarify that the technical proofs are included in the **Appendix of the main paper**, *not* in the Supplementary Material. Please refer to **Appendix D. Technical Proofs for the Main Paper** at the end of the main manuscript. To help readers locate the proofs more easily, we will explicitly highlight their locations in the final version of our manuscript:
> 1. The  proof of Theorems 4.1 can be found in Appendix D.*Proof of Theorem 4.1.* (Line 699);
> 2. The  proof of Theorems 4.2 can be found in Appendix D.*Proof of Theorem 4.2.* (Line 713);
> 3. The  proof of Propositions 6.2 \& 6.3 can be found in Appendix D.*Proof of Propositions 6.2 \& 6.3.* (Line 668).
>
> Lastly, we will clarify that  Lemma 6.1 was established in Leimer (1993), Page 105, Proposition 2.4. (iii).
>
> > **Weakness 2** *"Clarity of presentation of the evidence in Section 7 is recommended for a remake." "...It is suggested to use  relative measure over changing $r$ or vertices..."*
>
> Thanks for your comments. We have conducted additional experiments and will revise Section 7 to better demonstrate the advantage of the proposed method.
>
> We would like to clarify that the proposed algorithm (ICP) and the previous algorithm (CP) were tested over various specifications of **graph edge density** $r=|E|/|E_{\max}|$ with $|E_{\max}|=|V|(|V|-1)/2$.
>
> We performed experiments for $|V|=1024$ and $2048$, and the results are summarized in the tables below. As the edge density $r$ increases, the difference in running time (in seconds) between the two algorithms becomes more pronounced. This trend is attributed to the fact that denser graphs generally correspond to a lower value of $\theta=m/(|V|+|E|)$, which enhances the performance advantage of our proposed method.
>
> **Table1: Comparison between ICP and CP**
> |||||||
> |:-:|:-:|:-:|:-:|:-:|:-:|
> ||| $\vert V\vert=1024$ ||||
> |$r$|0.06|0.09|0.15|0.27|0.43|
> |$\mathrm{CP}-\mathrm{ICP}$ (seconds) |3.75|4.40|5.72|7.10|8.73|
> ||| $\vert V\vert=2048$ ||||
> |$r$|0.04|0.06|0.10|0.22|0.49|
> |$\mathrm{CP}-\mathrm{ICP}$ (seconds) |19.45 |34.47|41.54|59.24|62.24|
>
> Recall that our algorithm enhances Step 2 of Algorithm 1. When focusing solely on the computational cost of this step, the advantage of our method becomes even more evident. Let $T_{\mathrm{CP},2}$ be the average running time of Step 2 as implemented in Wienöbst et al. (2023), and let $T_{\mathrm{ICP},2}$ be the average running time of our improved approach for the same step. The table below presents both the difference and the ratio between these average running times. The results highlight substantial improvements achieved by our method in terms of efficiency.
>
> **Table2: Comparison between Step 2 of ICP and CP**
> |||||||
> |:-:|:-:|:-:|:-:|:-:|:-:|
> ||| $\vert V\vert=1024$ ||||
> |$r$|0.06|0.09|0.15|0.27|0.43|
> | $T_{\mathrm{CP},2}-T_{\mathrm{ICP},2}$ (seconds) | 3.31 | 4.65 |5.66 |8.04| 9.28 |
> |$T_{\mathrm{CP},2} / T_{\mathrm{ICP},2}$ |7.88|19.02|26.80|65.91|159.57|
> ||| $\vert V\vert=2048$ ||||
> |$r$|0.04|0.06|0.10|0.22|0.49|
> |$T_{\mathrm{CP},2}-T_{\mathrm{ICP},2}$ (seconds) |17.99| 24.19|37.41 |72.44 |81.45 |
> |$T_{\mathrm{CP},2} / T_{\mathrm{ICP},2}$ |10.05|16.42|29.71|92.31|318.54|
>
> The full new results will be included as new figures and tables in the revision. We believe these additional experiments will significantly enhance the evidence to support the advantages of the proposed method.
>
> > **Other Comments Or Suggestions:**
>
> Thank you for pointing this out. We will carefully proofread the paper and correct these typos in our final submission.

---

### Decision · Program_Chairs · 2025-05-01

**Decision:**

Accept (oral)

**Comment:**

All reviewers are unanimous in their opinion of this paper. This paper addresses an important algorithmic aspect of counting Markov equivalent (ME) DAGs efficiently by improving an existing algorithm. This ability to count ME DAGs is very useful for Bayesian approaches for experimental design. This paper makes a non trivial algorithmic contribution in this space with theoretical results backing it.

Recommendation: Accept